# Large-scale examination of early-age sex differences in neurotypical toddlers and those with autism spectrum disorder or other developmental conditions

Sanaz Nazari[1], Sara Ramos Cabo ®[1], Srinivasa Nalabolu[1], Cynthia Carter Barnes[1], Charlene Andreason[1], Javad Zahiri[1], Ahtziry Esquivel[1], Steven J. Arias[1], Andrea Grzybowski[1], Michael V. Lombardo[2], Linda Lopez[1], Eric Courchesne ®[1,3] ✉ & Karen Pierce ®[1,3] ✉

Autism spectrum disorder (ASD) is clinically heterogeneous, with ongoing debates about phenotypic differences between boys and girls. Understanding these differences, particularly at the age of first symptom onset, is critical for advancing early detection, uncovering aetiological mechanisms and improving interventions. Leveraging the Get SET Early programme, we analysed a cohort of 2,618 toddlers (mean age: ~27 months) through cross-sectional, longitudinal and clustering analyses, performed using statistical and machine learning approaches, to assess sex differences in groups with ASD, developmental delay and typical development across standardized and experimental measures, including eye tracking. The results revealed no significant sex differences in toddlers with ASD across 17 of 18 measures, including symptom severity based on the Autism Diagnostic Observation Schedule, receptive and expressive language based on the Mullen Scales of Early Learning and social attention based on the GeoPref eye-tracking test. In contrast, girls with typical development outperformed boys on several measures. Subtyping analyses stratifying toddlers into low, medium and high clusters similarly showed virtually no sex differences in ASD. Overall, our findings suggest that phenotypic sex differences are minimal or non-existent in those with ASD at the time of first symptom onset.

The importance of knowing whether there are clinical phenotypic differences between girls and boys with autism cannot be underestimated. Clarity on this topic can lead to improved early-age detection and diagnostic procedures, reveal new insights into causes and mechanisms, point to more efficacious early-age treatment protocols, aid in differential planning and targets for intervention and more generally provide enhanced sex and gender equity in society[1]. Examination of early-age sex differences within the context of

longitudinal data in particular can generate important insights into sex-specific developmental trajectories, benefiting parents and clinicians in understanding progression and course. Inconsistent and contradictory reports on sex differences in autism are the source of debate inside and outside the scientific and clinical communities. Some studies report symptom differences[2,3] and some do not[4]; some report cognitive differences[5] and some do not[6]; some report female versus male differences in autism spectrum disorder (ASD) that are similar

to sex differences in those who are typically developing (TD)[7] and some do not[4].

For example, females with ASD have been reported to have superior social skills[8,9], subtler autistic traits[10], fewer restricted and repetitive behaviours (RRB)[2,11,12] and superior language abilities compared with males with ASD[13]. In contrast, other studies have reported that girls with autism have more social difficulties[6,14,15] and display more intellectual disability than males with autism[16]. Yet, still other studies report that females and males with ASD are similar in cognition[17,18], attention[19] and behavioural traits associated with ASD[20,21]. Thus, there is currently a pronounced lack of clarity on the question of clinical phenotypic differences between girls and boys with autism, and this has led to widely divergent causal and developmental theories, such as the protective effect theory[22,23] and the extreme male brain (EMB) theory[24]. This has also led to concerns about equity in detecting and diagnosing ASD in girls and providing clinical care for girls with ASD[25]. This situation may be less due to the biological and/or clinical complexity of sex and autism factors and more due to study design and small sample limitations.

There is now considerable evidence suggesting that ASD begins during prenatal life[26–28]. Within this context, examination of sex differences at the earliest ages possible is essential given the large impact of very early experience on phenotypic expression[29]. For example, in one study, changes in the quality of the home environment dramatically influenced cognitive profiles in infants originally placed in an orphanage by as much as 15 IQ points[30]. In another study, the receptive language ability of children with autism was considerably improved if exposure to a particular intervention occurred at or before 18 months of age[31]. Although studying extremely young individuals does not entirely eliminate the confound of experience, it provides a snapshot of ASD more proximal to the disorder's onset, which may generate results that are more biologically, and less experientially, driven. In contrast, studies of older children and adults, while important for understanding the condition and complimentary to studies with infants and toddlers, cannot necessarily disentangle the influence of experience. However, more sex differences are reported in studies involving older children[3,10,32,33], highlighting the importance of investigating sex differences at early ages.

Overall, limitations in previous studies span a wide range. The great majority of studies have been small samples of $n = 28–96$ and thus lack statistical power[6,7]. Very few studies examined the earliest ages possible, when effects are closest to autism prenatal beginnings[27,28,34] and before a host of experiential effects could come into play. Ascertainment bias weakens the generalizability of sex difference findings. For instance, ascertainment from clinic-referred samples may be biased towards those who are more cognitively and socially affected, as in a recent study where 98% of toddlers scored in the cognitively impaired range with an early learning composite (ELC) score of <70 (ref. 17). Similarly, the ascertainment of infant siblings of children with autism from within multiplex families may also be unrepresentative of the general ASD population[4,5]. Assessment limitations provide incomplete or weakly validated clinical phenotypic information about social symptom severity, language ability, cognitive level and social attention behaviour. For instance, some studies used parent report rather than gold-standard expert-based assessment tools[35] and some reported cognitive but not ASD symptom test results[17] and vice versa[3]. Few examined longitudinal changes in sex effects[36,37]. Some studies lack comparison with TD participants and/or participants with developmental delay (DD), thus limiting the interpretation of sex differences and sex-effect specificity in ASD[12,18]. Lastly, nearly all studies lacked unsupervised patient subtyping, which enables data-driven analyses of phenotypes in girls and boys with ASD across different levels of clinical ability from higher to lower.

We utilized our general San Diego County-wide population-based screening approach, Get SET Early[38], to screen, evaluate and identify a large cohort of females and males with early-age ASD. This method generated a large single-site study sample containing $n = 2,618$ toddlers (1,539 with ASD and 1,079 without ASD), including toddlers with ASD as young as 12 months of age (Table 1). Importantly, the Get SET Early method simultaneously enabled us to recruit, examine and assess in a comparable and unbiased way TD as well as non-ASD developmental delayed (DD) girls and boys for comparison to the ASD toddlers. We further leveraged data integration and machine learning approaches to examine heterogeneity in males and females with ASD, utilized a social attention eye-tracking test and leveraged longitudinal data to examine sex-specific early-age psychometric and social symptom developmental changes associated with ASD.

## Results

### Primary analysis

Two-way analyses of variance (ANOVAs) across all subscales revealed significant omnibus $F$-test results for the entire model (Supplementary Table 1) and we examined our hypotheses when appropriate (see Methods for detailed steps and procedures).

**Sex differences in ASD.** Although the parent-based screening tool used to recruit toddlers to the study (that is, the Communication and Symbolic Behavior Scales (CSBS) Infant-Toddler Checklist[39]) revealed lower screen scores in boys with ASD than girls with ASD, we found no significant sex differences in 17 of the 18 standardized test scores presented in Fig. 1. Similarly, on the GeoPref eye-tracking test, no sex difference among girls and boys was found, as reported in our earlier paper[40] (Fig. 2a). Only for the Vineland Adaptive Behavior Scales (Vineland)[41] daily living skills subdomain did girls with ASD score significantly higher than boys with ASD (Extended Data Table 1).

**Sex differences in TD.** Similar to toddlers with ASD, on the parent-based CSBS checklist, TD boys had poorer screen scores than girls on social, symbolic and total score composites. However, in contrast with toddlers with ASD, ten of the 18 standardized test comparisons in Fig. 1 showed significant sex differences, with TD girls performing better than TD boys. On the Autism Diagnostic Observation Schedule (ADOS)[42], in the TD group girls had slightly better scores than boys. On the Mullen Scales of Early Learning (MSEL) test, TD girls had higher IQ scores than TD boys for fine motor skills, visual reception and receptive language. Similarly, on the MacArthur-Bates Communicative Development Inventories (CDI)[43] words and gestures (WG) analysis, TD girls had higher scores on later gestures and total gestures. On the CDI words and sentences (WS) analysis, TD girls also had higher scores on the words produced subscale. On the GeoPref eye-tracking test, there were no sex difference among TD girls and boys. In addition, on the Vineland parent questionnaire, TD girls scored better than TD boys for daily living skills, as well as the adaptive behaviour composite (ABC) and socialization (Extended Data Table 1).

It is noteworthy that the ANOVA results indicated significant interactions between sex and group for only four subscales: ADOS social affect, MSEL visual reception, MSEL receptive language and CDI–WS words produced. These interaction effects can be described as subtle but significant sex differences in the TD group, with a female advantage over males, but a lack of sex differences in the group with ASD. For the remaining subscales showing statistically significant differences, these differences were driven by the significant main effect of sex. Therefore, sex differences in the ASD and TD groups should be interpreted independently for these other measures (Fig. 2b).

**Sex differences in DD.** There were almost no sex differences between girls and boys with DD. Girls showed slightly better performance than boys with an ignorable effect size in ADOS RRB and girls scored slightly better on Vineland motor skills, with a small effect size (Extended Data Table 2).

**Table 1 | Demographics and clinical characteristics summary**

| | ASD (F) | ASD (M) | TD (F) | TD (M) | DD (F) | DD (M) |
|---|---|---|---|---|---|---|
| *n* | 339 | 1,200 | 252 | 349 | 129 | 349 |
| **Ethnicity** | | | | | | |
| Hispanic/Latino (%) | 40.1 | 37.2 | 20.6 | 20.6 | 45.7 | 46.4 |
| Not Hispanic/Latino (%) | 53.1 | 55.9 | 70.6 | 73.1 | 48.1 | 47.6 |
| Unknown/not reported (%) | 6.8 | 6.9 | 8.7 | 6.3 | 6.2 | 6.0 |
| **Race** | | | | | | |
| American Indian/Alaska Native (%) | 1.8 | 0.8 | 0 | 0.6 | 0 | 2.6 |
| Asian (%) | 14.2 | 13.8 | 6.7 | 13.2 | 10.1 | 6.6 |
| Black/African American (%) | 4.4 | 2.2 | 2.0 | 3.2 | 2.3 | 4.3 |
| More than one race (%) | 10.3 | 12.7 | 7.9 | 7.7 | 5.4 | 10.0 |
| Native Hawaiian/other Pacific Islander (%) | 1.2 | 1.3 | 1.2 | 1.1 | 0.8 | 2.0 |
| Unknown/not reported (%) | 18.6 | 17.6 | 13.9 | 9.7 | 22.5 | 22.9 |
| White (%) | 49.6 | 51.7 | 68.3 | 64.5 | 58.9 | 51.6 |
| **SES (median household income)[a] (US$)** | 94,413 | 99,057 | 103,122 | 109,313 | 92,299 | 89,577 |
| **Longitudinal analysis** | | | | | | |
| **ADOS (*n*)** | 341 | 1,198 | 253 | 346 | | |
| Mean age±s.d. (range) (months) | 28.1±8.0 (12.0–48.1) | 29.3±8.2 (12.0–48.4) | 26.0±10.1 (12.1–48.2) | 25.2±10.0 (12.0–48.5) | | |
| **Vineland (*n*)** | 339 | 1,200 | 252 | 349 | | |
| Mean age±s.d. (range) (months) | 27.9±8.2 (12.0–48.1) | 28.8±8.1 (12.0–48.4) | 25.4±10.1 (12.0–48.2) | 25.1±10.1 (12.0–48.4) | | |
| **MSEL (*n*)** | 339 | 1,192 | 239 | 338 | | |
| Mean age±s.d. (range) (months) | 28.1±8.2 (12.0–48.1) | 29.2±8.2 (12.0–48.4) | 23.3±9.1 (12.1–48.2) | 23.7±9.1 (11.9–48.5) | | |
| **Primary analysis** | | | | | | |
| **ADOS (*n*)** | 282 | 924 | 253 | 346 | 129 | 346 |
| Mean age±s.d. (range) (months) | 25.7±6.7 (12.0–48.1) | 26.2±6.5 (12.0–47.2) | 26.0±10.1 (12.1–48.2) | 25.2±10.0 (12.0–48.5) | 28.9±9.9 (12.2–46.4) | 27.9±9.4 (12.1–47.2) |
| **Vineland (*n*)** | 273 | 941 | 252 | 349 | 129 | 349 |
| Mean age±s.d. (range) (months) | 25.4±6.7 (12.0–48.1) | 26.1±6.6 (12.0–47.2) | 25.4±10.1 (12.0–48.2) | 25.1±10.1 (12.0–48.4) | 28.7±9.9 (12.0–46.4) | 27.8±9.5 (12.1–47.2) |
| **MSEL (*n*)** | 207 | 667 | 239 | 338 | 130 | 347 |
| Mean age±s.d. (range) (months) | 23.8±6.8 (12.0–48.1) | 24.2±6.8 (12.0–47.2) | 23.3±9.1 (12.1–48.2) | 23.7±9.1 (11.9–48.5) | 28.7±9.9 (12.2–46.4) | 27.9±9.5 (12.1–47.2) |
| **CDI–WG (*n*)** | 55 | 176 | 98 | 136 | 21 | 63 |
| Mean age±s.d. (range) (months) | 15.2±1.7 (12.1–17.7) | 14.9±1.7 (9.6–18.5) | 14.9±1.6 (12.5–18.5) | 14.7±1.6 (9.6–18.5) | 14.9±1.8 (12.2–18.3) | 15.1±1.7 (12.1–18.4) |
| **CDI–WS (*n*)** | 141 | 482 | 108 | 161 | 13 | 69 |
| Mean age±s.d. (range) (months) | 24.4±3.1 (18.0–30.5) | 24.8±3.3 (15.6–30.5) | 25.1±3.4 (17.9–30.5) | 24.5±3.6 (16.4–30.5) | 24.9±2.4 (21.1–28.9) | 24.2±3.5 (18.6–30.1) |
| **CSBS (*n*)** | 96 | 361 | 183 | 274 | 44 | 120 |
| Mean age±s.d. (range) (months) | 16.2±3.6 (11–24) | 16.2±4.0 (9–24) | 16.2±4.2 (10–24) | 15.9±4.0 (9–24) | 17.6±3.6 (12–24) | 17.7±3.5 (12–24) |
| **GeoPref (*n*)** | 131 | 397 | 185 | 255 | 72 | 219 |
| Mean age±s.d. (range) (months) | 27.3±10.0 (12.3–48.1) | 27.2±9.8 (12.2–47.3) | 26.6±10.3 (11.9–48.1) | 26.8±9.7 (12.0–48.5) | 26.5±9.2 (12.2–46.4) | 27.5±9.5 (12.8–47.2) |
| **Cluster analysis** | | | | | | |
| *n* | 254 | 862 | 235 | 323 | | |
| **ADOS** | | | | | | |
| Mean age±s.d. (range) (months) | 25.5±6.8 (12.6–48.1) | 26.0±7.0 (12.6–47.2) | 25.5±9.0 (12.1–48.2) | 26.89±9.0 (12.0–48.5) | | |

**Table 1 (continued) | Demographics and clinical characteristics summary**

| | ASD (F) | ASD (M) | TD (F) | TD (M) | DD (F) | DD (M) |
|---|---|---|---|---|---|---|
| **Vineland** | | | | | | |
| Mean age±s.d. (range) (months) | 25.2±6.8 (12.0–48.1) | 25.7±7.0 (12.0–47.2) | 25.3±9.0 (12.0–48.2) | 26.7±9.0 (12.0–48.5) | | |
| **MSEL** | | | | | | |
| Mean age±s.d. (range) (months) | 25.4±6.8 (12.5–48.1) | 25.9±7.0 (12.4–47.2) | 25.4±9.1 (12.1–48.2) | 26.8±9.1 (12.0–48.5) | | |

ªThe socioeconomic status (SES, as determined by household income) for 30 participants was not reported. Sample sizes may vary across tests due to differences in age ranges and/or the presence of missing data.

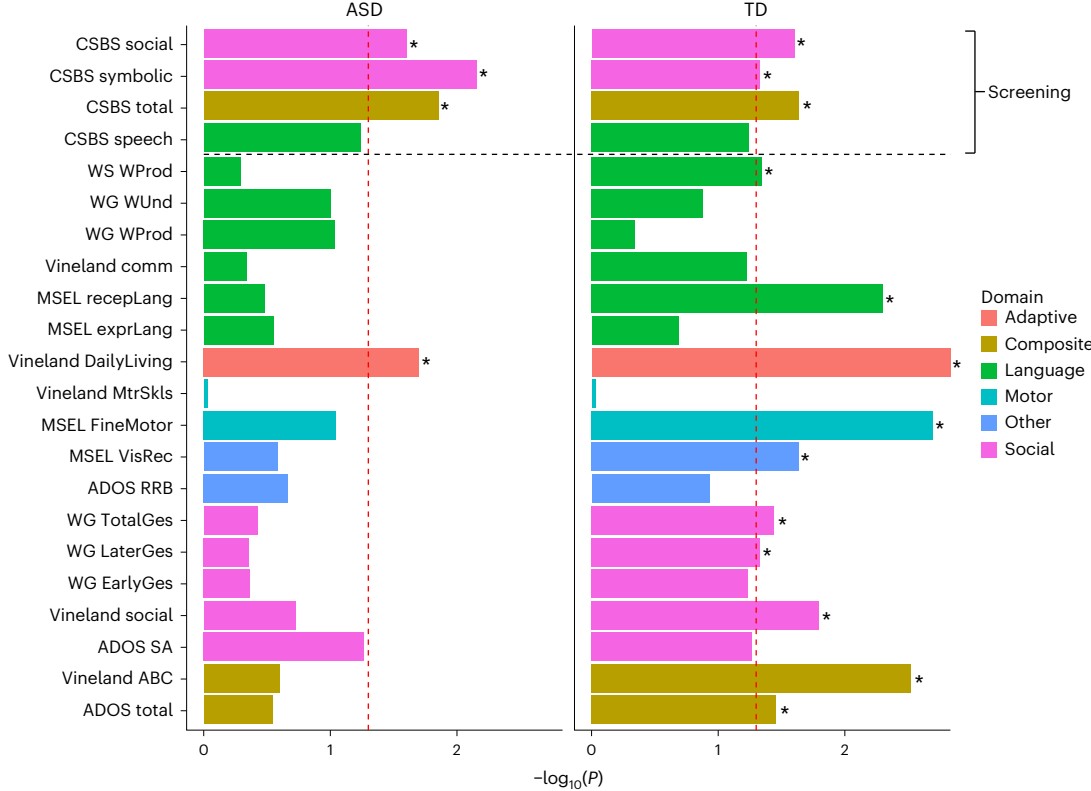

**Fig. 1 | Comparison of sex differences in ASD and TD groups across various test subscales.** The bars represent performance differences between males and females, with longer bars indicating stronger statistical significance (lower P values). Any bar crossing the red dashed line has a P value of <0.05. Asterisks denote significant sex differences. In all cases where a significant difference is marked, girls outperformed boys. For the ADOS, where lower scores indicate less impairment, outperformance means lower scores for girls. For the Vineland scales, where higher scores represent better abilities, outperformance means higher scores for girls. All of the tests used to determine significance were two sided. Either a Kruskal–Wallis chi-squared test or a t-test was used, as appropriate (Extended Data Table 1). Multiple comparisons were corrected for using the false discovery rate. comm, communication; DailyLiving, daily living skills; EarlyGes, early gestures; exprLang, expressive language; FineMotor, fine motor skills; LaterGes, later gestures; MtrSkls, motor skills; recepLang, receptive language; SA, social affect; TotalGes, total gestures; VisRec, visual reception; WProd, words produced; WUnd, words understood.

## Cluster analysis

After reviewing the results of 26 techniques to determine the number of clusters (Supplementary Table 2), there was a tie between two- and three-cluster solutions. We proceeded with analyses of three clusters. Based on our experience of clustering children with ASD and TD toddlers using similarity network fusion (SNF), the first obvious result of SNF would be a two-cluster solution, as children with ASD and TD toddlers tend to separate easily based on their scores. However, in this study, we opted for a three-cluster solution, which resulted in high, medium and low clusters, two of which contained both individuals with ASD and TD toddlers. This approach allowed us to capture a broader range of heterogeneity within the sample, providing more detailed and nuanced insights into the varying characteristics and different levels of traits. The three clusters resulted in toddlers with high, medium and low

performance in the social, language and motor domains. Toddlers with ASD spanned all three cluster performance levels, but TD toddlers were only present in the high and medium performance clusters, as expected.

**Validation of clusters.** We trained SNF[44] on 80% of the data (1,337 participants) and tested it on the remaining 20% (336 participants) (Methods). The three clusters were consistently observed in both the training and test datasets (Fig. 3). To validate separation of the clusters and assess differences between them, we conducted ANOVAs and multiple pairwise comparisons on the training and test clusters. The results showed a significant omnibus ANOVA and significant pairwise comparisons across all three clusters, spanning the variables used to construct the SNF across the three data layers. Additionally, in predicting the test data clusters using the trained SNF model, we

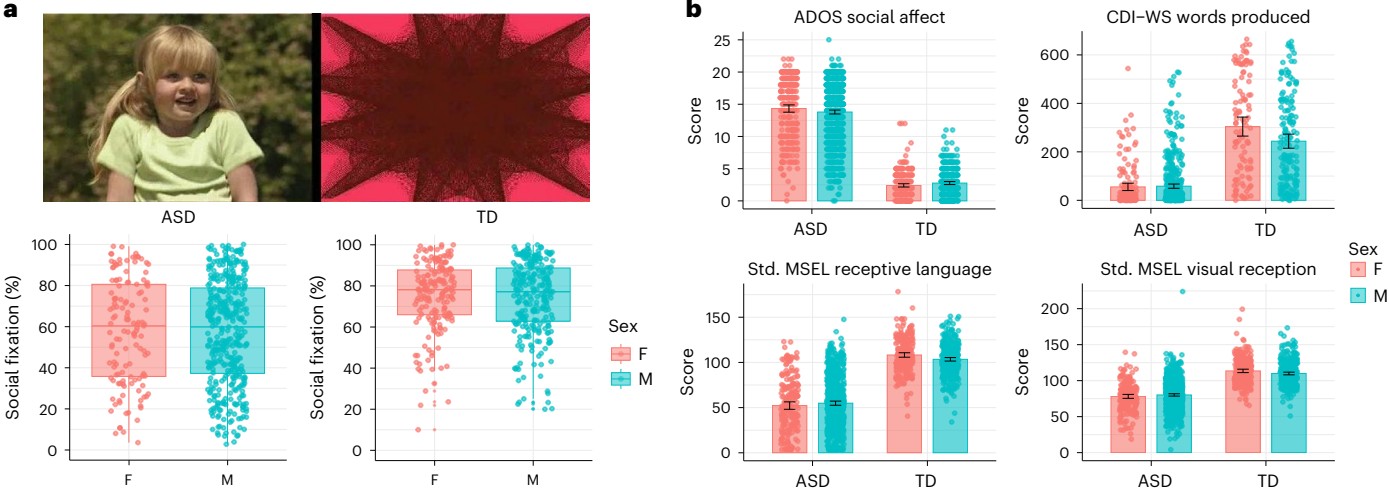

**Fig. 2 | Primary analysis of sex differences in ASD and TD groups. a**, Top, images from the GeoPref eye-tracking test—a tool that analyses visual stimulus preferences in toddlers. Bottom, bar graphs showing that there were no statistically significant sex differences in the results of this test in the group with ASD or the TD group (Extended Data Table 1). Each box plot illustrates the data distribution, with the centre line representing the median, the box edges indicating the interquartile range (IQR; 25th–75th percentiles) and the whiskers extending to the smallest and largest values within 1.5× the IQR. **b**, Test subscales that exhibited statistically significant interactions between sex and diagnostic group, with the TD group showing sex differences (Supplementary Table 1). Mullen subscales show standardized (Std.) age-equivalent scores. The data are presented as mean values ± CIs; see the primary analysis section of Table 1 for sample sizes. Please see the datasets df.match.geopref.csv (**a**) and df.match. ados2.csv, df.match.mul2.csv, df.match.wg.csv and unq.ws2.csv (**b**) in GitHub (Primary Cross-Sectional Analysis→Data; https://github.com/ACE-UCSD/Autism-Sex-Differences-Analysis-Pathway). Images in panel **a** © 2003 Gaiam Americas, Inc., courtesy of Gaiam Americas, Inc. and Fit For Life, LLC.

observed a similar structure to that of the training data. This included toddlers with ASD in the high-, medium- and low-ability clusters and TD toddlers in the high- and medium-ability clusters. Furthermore, the proportions of the test sample assigned to the clusters were relatively comparable to those of the trained model (Supplementary Tables 3–6). We computed silhouette scores for both the training and test data clusters to further validate the cluster separation. The results showed overall high scores of 0.46 and 0.40 for the training and test clusters, respectively, indicating distinct and well-separated clusters (Fig. 4 and Supplementary Table 7).

To examine the SNF clusters in relation to external variables, we compared the clusters for WG words produced, words understood, early gestures, later gestures, total gestures, WS words produced, ADOS RRB and the percentage of social fixation in the eye-tracking GeoPref test. The ANOVA results revealed that at least two of the three cluster comparisons showed significant differences across all external variables. This was consistent with the clear separation observed in the variables used to construct the SNF clusters, for which all three pairwise comparisons were significant (Fig. 4 and Supplementary Tables 8 and 9). Furthermore, performing a fivefold cross-validation on the training set, repeated ten times, resulted in a high average accuracy of 0.91. Finally, robustness analyses demonstrated the high stability of SNF against data perturbations. When subjected to random removal of 5, 10, 20, 30, 40 and 50% of the study participants, SNF exhibited high average normalized mutual information (NMI) values of 90.8, 88.3, 84.5, 81.3, 78.6 and 75.2%, respectively (Fig. 4).

Next, we tested whether sex differences varied across the heterogeneous high, medium and low spectrum of clinical performance seen in the SNF clusters.

**Sex differences in those with ASD within and across clusters.** There were few differences between girls and boys with ASD within and across clusters. In the training SNF data, girls with ASD in the medium cluster had worse ADOS social affect scores than boys with ASD, yet better socialization scores on the Vineland parent questionnaire. No

other significant sex differences were found for the ASD group in the high-, medium- and low-ability clusters in the training datasets. Those few sex differences in the medium-ability cluster were not replicated in the test SNF data (for example, social affect scores were 14.5 and 14.1 in girls and boys, respectively, and not different). Instead, in the test dataset girls with ASD in the medium cluster had better MSEL fine motor scores than boys with ASD in this cluster, and girls with ASD in the high-ability cluster had higher Vineland communication and MSEL expressive language scores than boys with ASD in this cluster. These several better scores in girls with ASD were not seen in the larger-sample training dataset (Extended Data Tables 3 and 4 and Supplementary Tables 10 and 11).

We also investigated sex effects on external variables and found no significant sex differences in the high- and medium-performing clusters for the group with ASD. However, in the low-performing group, males with ASD scored higher for WS words produced. It is noteworthy that in the low-ability cluster for the group with ASD, we did not have enough observations for girls to examine sex differences in several external variables (Extended Data Table 5 and Supplementary Table 12).

**Sex differences in TD toddlers within and across clusters.** Among the clusters, there were several sex differences for TD toddlers. In the training SNF dataset, among higher-ability toddlers, TD girls achieved better scores than boys on Vineland socialization, Vineland motor skills, MSEL fine motor skills and MSEL receptive language; however, these sex differences did not replicate in the smaller sample of higher-ability TD toddlers in the test dataset. Similarly, TD boys scored higher than TD girls on MSEL expressive language in the medium-ability cluster and these differences were not observed in the smaller testing dataset (Extended Data Tables 3 and 4 and Supplementary Tables 10 and 11).

We also investigated sex effects on external variables within the TD group. Girls in the high-ability cluster had better scores on WG early gestures, total gestures and WS words produced than boys; and girls in the medium-ability cluster had better ADOS RRB scores than boys.

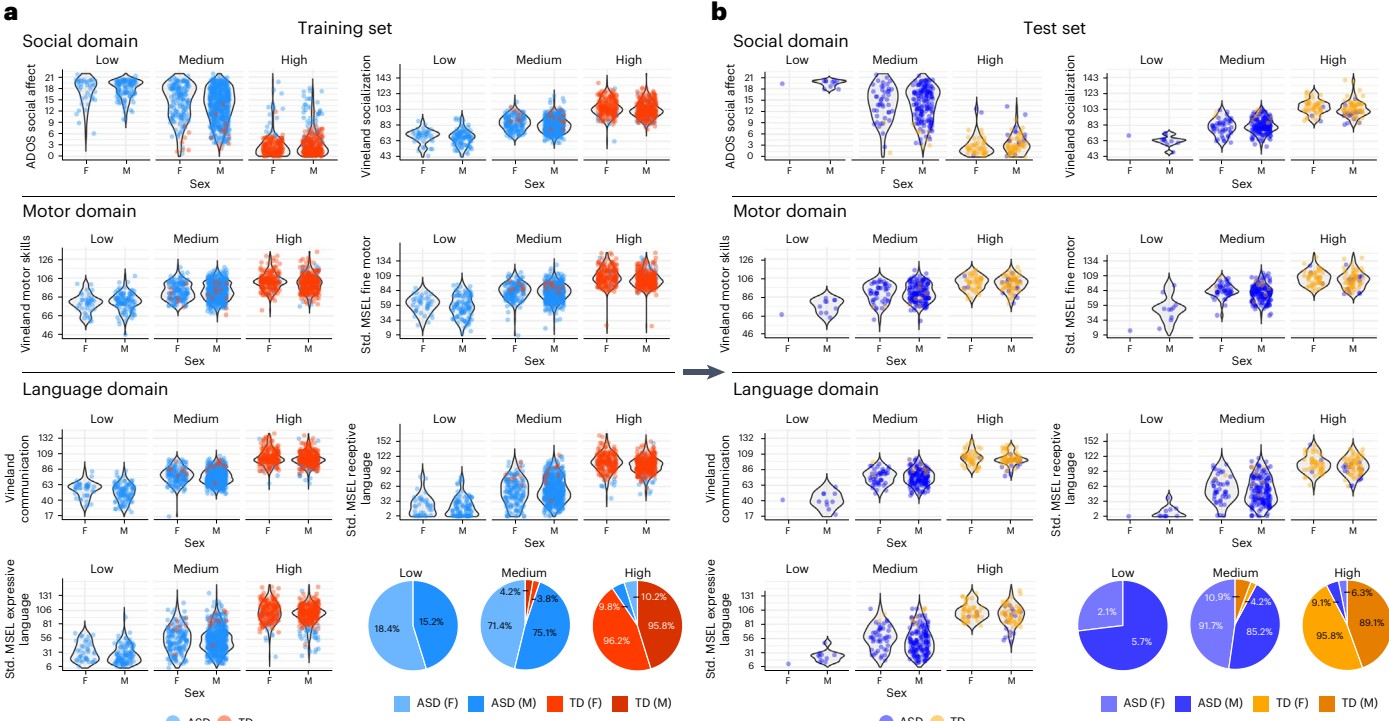

**Fig. 3 | Cluster analysis. a,b,** Graphs for the training set (**a**) and test set (**b**) illustrating the separation of high, medium and low clusters across three domains: social, motor and language (Supplementary Tables 3–6). For each domain, violin plots display the density and distribution of scores for the group with ASD and TD toddlers, separated by sex. Individual data points are overlaid on the violin plots to highlight the variability within each cluster. The social domain includes the measures ADOS social affect and Vineland socialization; the motor domain features Vineland motor skills and MSEL fine motor skills;

and the language domain comprises Vineland communication, MSEL receptive language and MSEL expressive language. The pie charts show the proportional distribution of ASD and TD groups across clusters and sexes, emphasizing the distinct separation and composition of the clusters. The clustering patterns are consistent between the training and testing sets, validating the robustness of the clustering approach across all domains. Please see the datasets train.labels.asd. td2.csv and test.labels.asd.td.csv in GitHub (Cluster Analysis; https://github.com/ACE-UCSD/Autism-Sex-Differences-Analysis-Pathway).

These better scores in different domains in TD girls generally align with the results obtained from the trained SNF model (Extended Data Table 5 and Supplementary Table 12).

To examine sex differences across clusters, the results of a chi-squared test revealed no significant difference in the distribution of boys and girls across the three clusters in both those with ASD and TD toddlers in the training data, indicating that sex was not a determining factor in cluster membership (Supplementary Table 13).

**Longitudinal analysis**
Consistent with the primary analysis results, longitudinal analysis revealed more and stronger sex differences in TD toddlers than in those with ASD. These sex differences in TD toddlers occurred in both initial status (that is, intercept) and growth trajectories (that is, slope) and the estimates in Extended Data Table 6 indicate the difference between boys' and girls' scores regarding their intercept and slope.

**Longitudinal sex differences in those with ASD.** On the ADOS, boys and girls with ASD did not differ in intercepts. However, boys with ASD exhibited age-related ADOS social affect and total score growth trajectories that were increasingly worse and became nearly identical to those of girls by later toddler ages (slopes in Extended Data Table 6), and effect sizes were small (that is, 0.42 and 0.44, respectively). On the MSEL, no significant longitudinal sex differences among toddlers with ASD were found. On the Vineland parent questionnaire, girls with ASD had a slightly but significantly less declining slope than boys with ASD for the motor skills subscale, but again the effect size was small (0.43; Fig. 5).

**Longitudinal sex differences in the TD group.** On the ADOS, TD girls had significantly better initial social affect, RRB and total scores than TD boys. Compared with TD boys, girls in this group displayed better longitudinal improvement in social affect and total scores (effect sizes ranged from small to medium: 0.48–0.79) (Extended Data Table 6 and Fig. 5). On the MSEL, TD girls exhibited significantly better initial scores than boys for the visual reception, receptive language and expressive language subscales by 4.12, 6.82 and 3.73, with effect sizes of 0.43, 0.72 and 0.36, respectively. On the Vineland parent questionnaire, TD girls once again had better intercept scores than boys for communication, daily living skills, socialization and ABC, with differences of 2.76, 3.78, 2.07 and 2.46, respectively. However, the effect sizes were small at 0.35, 0.49, 0.30 and 0.36, respectively. Additionally, on the Vineland motor skills subscale, TD girls had better improvement in longitudinal trajectory than TD boys, with an average slope difference of 0.16 and a medium effect size of 0.55 (Fig. 5).

**Interpretation of confidence intervals and precision**
The non-significant findings reported in this study are supported by the precision of the estimates, as reflected in the calculated confidence intervals (CIs). In the primary analyses, narrow CIs for the mean or median differences across groups suggest that the observed null effects are unlikely to be due to insufficient power or variability in the data. Instead, these intervals indicate that any potential differences, if present, are probably negligible and not clinically meaningful. Similarly, in the longitudinal analyses, the CIs for both intercepts and slopes in the latent growth models demonstrate high precision in estimating baseline levels and rates of change over time. These precise estimates reinforce the robustness of the null results and the consistency of the

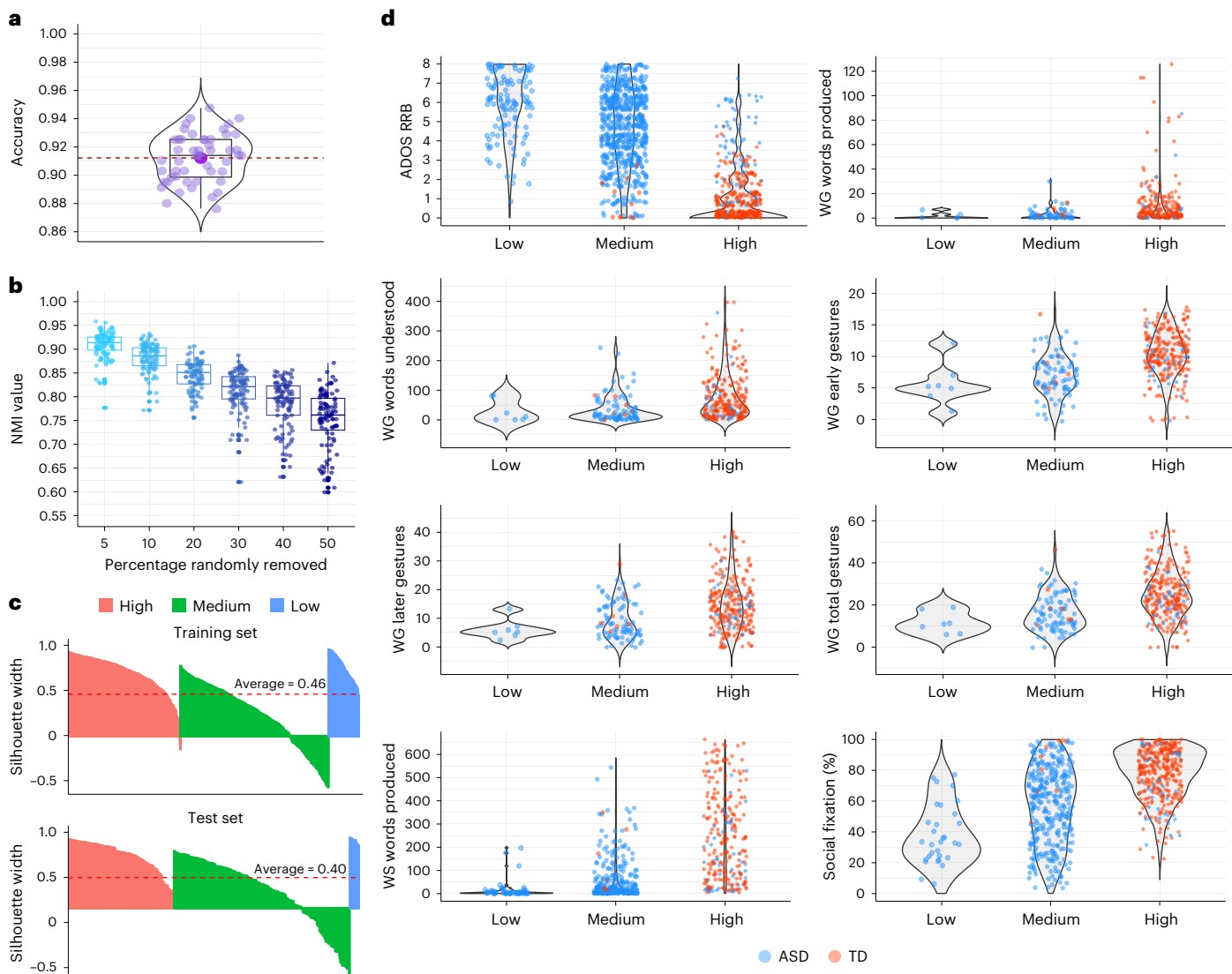

**Fig. 4 | Cluster validation strategies. a**, Results of fivefold cross-validation on the training set, repeated ten times, yielding a high average accuracy of 0.91. The embedded box plot highlights key summary statistics: the median (centre line), IQR (box boundaries) and values within 1.5× the IQR (whiskers). Individual data points show accuracy values. The mean accuracy is indicated by the large purple point, with the dashed red line representing the overall mean accuracy across all runs. **b**, Graph showing high NMI values remaining consistent across increasing percentages of random removals. Each box plot represents the distribution of NMI values from 100 replicates per removal percentage, with the centre line indicating the median, the box boundaries showing the IQR (25th–75th percentile) and the whiskers extending to 1.5× the IQR. **c**, Confirmation of the quality of clustering in both the training and test sets, as evidenced by silhouette scores of 0.46 and 0.40, respectively. **d**, Graphs showing the distinct separation of clusters when external variables are applied (Supplementary Tables 8 and 9).

findings across multiple time points. The absence of wide CIs across analyses further highlights the stability of the results and provides strong evidence that sex differences in ASD at early ages are minimal or non-existent.

## Discussion

Our large-scale study shows that at the early-age clinical beginning of autism, there are virtually no clinically or statistically significant differences between female and male toddlers with ASD across a wide range of standardized and validated tests of symptom severity, social and language ability and behavioural social attention. This study included *n* = 2,618 contrast male and female toddlers with ASD and DD and TD individuals from the general San Diego population, with the majority uniformly ascertained and recruited using the Get SET Early model[38]. Toddlers were psychometrically and diagnostically assessed at a single site by licensed psychologists, thus the participants were not a

collection from different sites with varying procedures, personnel and populations. Only one of 19 primary study test comparisons between female and male toddlers with ASD was significantly different. This single difference was within the daily living subdomain score on a parent report tool—the Vineland—with girls scoring higher than boys. In contrast, ten of 18 measures were significantly different between typically developing females and males, with female toddlers consistently performing better than male toddlers. Furthermore, the lack of sex differences within the group with ASD was not unique to those with ASD since we also found almost no sex differences in toddlers with DD who did not have ASD.

Our findings—that there are no clinical sex differences in ASD at very early ages—leads to two possible conclusions. The first is that previous studies that reported clinical sex differences in ASD are incorrect, possibly due to small sample sizes, sampling bias, limited study measures or other methodological issues. An alternative conclusion

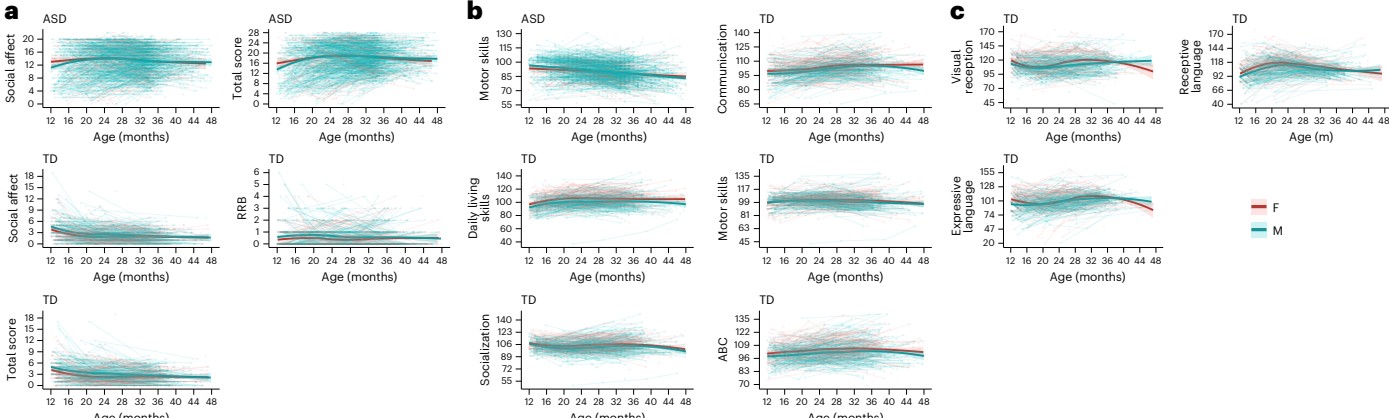

**Fig. 5 | Longitudinal analysis. a–c,** Longitudinal growth trajectories and baseline differences across the ADOS (**a**), Vineland (**b**) and MSEL subscales (**c**), stratified by sex. Only subscales with statistically significant sex differences in intercept and/or slope (*P* < 0.05) are shown (Extended Data Table 6). For the ADOS, significant differences were observed in the intercept and slope for TD-Social Affect, the slope for ASD-Social Affect, the intercept for TD-RRB, the intercept and slope for TD-Overall Total, and the slope for ASD-Overall Total. For the Vineland, significant differences were found in the intercept for TD-Communication, TD-Daily Living Skills, TD-Socialization and TD-ABC, and

in the slope for ASD-Motor Skills and TD-Motor Skills. For the MSEL, significant intercept differences were identified for TD-Visual Reception, TD-Receptive Language and TD-Expressive Language. Smoothed trend lines represent estimated values with 95% CIs (shaded areas) and individual participant trajectories are shown as thin lines to highlight variability. See the datasets ados.long.csv, vine.long.csv and mul.long.csv in the GitHub repository (Longitudinal Analysis; https://github.com/ACE-UCSD/Autism-Sex-Differences-Analysis-Pathway).

is that sex differences do not exist in ASD at the time of first symptom onset, but emerge slowly at later ages, driven by psychosocial factors or differences in biology between males and females that unfold across development. At the psychosocial level, studies have shown that parents engage in more positive parenting behaviour with their female children relative to males, which can lead to sex differences in language expression[45]. At the biological level, differences in sex hormone surges can also impact clinical phenotype[46]. Longitudinal studies that track children for whom ASD was detected early through to school age and beyond and then compare early- versus later-age symptom presentation could help to resolve this question.

Similar to other studies, we found stronger performance for female TD toddlers than male TD toddlers. For example, studies on neurotypical development note that female TD toddlers display better social and linguistic abilities, as well as increased attention to faces[47] and greater eye contact[48], relative to male toddlers. Moreover, typical girls produce more gestures and more words than boys[49–51]—a phenomenon that has been reported across languages and cultures[52]. Lange et al.[53] showed that typical girls have larger vocabularies, and preschool girls in particular (3–6 years of age) have better grammar, speech comprehension, pronunciation and processing of sentences and nonce words. In the present study, females with ASD did not show such differences compared with males with ASD. Although the study was not designed to test the EMB theory of ASD[24], the results—showing a similar female over male advantage in the TD group but no sex difference in the group with ASD for the ADOS social affect scale—align with EMB predictions. The EMB theory posits that in domains with typical sex differences (for example, early social and language abilities), these differences would be attenuated in ASD, with ASD presenting at the extreme end of the spectrum compared with TD males[24].

Clinical heterogeneity in ASD is well known, but there are few large-sample studies addressing whether or not there are sex differences in clinical heterogeneity at very early ages. Here we considered this topic using state-of-precision-medicine methods to determine subtype membership for study toddlers, and leveraged comprehensive validation strategies including cluster separation analyses, fivefold cross-validation, hold-out test set analysis, robustness assessment, cluster quality assessment and external validation analyses[54]. Using SNF—a robust, unsupervised, multimodality integration approach—we

identified three reproducible subtypes of toddler with ASD, best described as low, medium and high ability. The profile of the low-ability ASD subtype aligns well with the clinical characteristics of profound autism[55], whereas the high-ability ASD subtype overlaps with typical toddlers, and many patients assigned to the high-ability subtype may go on to have optimal later-age outcomes. The heterogeneity and subtypes in the present study represent the wide spectrum known for autism, ranging from severely affected to high functioning. Ninety percent of patients with ASD fell into the low- and medium-ability clusters, whereas 96% of TD toddlers fell into the high-ability cluster. Thus, SNF is an accurate unsupervised multimodality approach for ASD versus TD diagnostic separation and clinical subtyping. Within each of the three subtypes of this ASD spectrum, girls and boys with ASD in our study had remarkably similar symptom severity and social, language, cognitive and social eye-tracking levels of performance. In addition, cluster analyses revealed that the proportions of girls and boys with ASD were the same at low, medium and high levels of ability. Thus, girls with ASD were not disproportionately impaired relative to boys, nor were they disproportionately higher functioning. Although ASD is highly heritable[56] and girls are thought to carry higher genetic liability, leading to them potentially being more impacted[22], the lack of clinical differences in ASD between girls and boys at the early onset of the disorder does not fit that hypothesis.

In this study, we not only examined sex differences at the very young age of first symptom expression, but also possible changes across time. Our results did not reveal striking differences in trajectories between male and female children with ASD. Given the considerable biological heterogeneity in ASD, however, we hypothesize that subtype-specific hypotheses may be more informative than sex[57].

Although the present study had numerous strengths, including a large sample size (that is, 2,618 toddlers) and an extensive test battery with both standardized and eye-tracking measures, there are two potential limitations. First, although data collection at a single site using standardized operating procedures and licensed clinical psychologists probably minimized the noise that is often associated with multi-site studies, it is unclear whether the results would generalize to other geographical regions. Second, it is important to consider the possible influence of our approach to recruiting participants with early-detected ASD (that is, Get SET Early, which relies on a toddler's

failure of the CSBS screening at well-baby check-ups to prompt a referral to our centre) on the study results. Although sex-specific norms are not provided by the test developers, it is possible that this screening method is less sensitive to the detection of autism in younger girls given that male toddlers with ASD had slightly lower (worse) CSBS scores than females with ASD. However, counter to this is the fact that the male:female ratio in the current study was actually more strongly in favour of the detection of ASD in females than the national average (3.6:1.0 in the current study versus 3.8:1.0 in the most recent Centers for Disease Control and Prevention average[58]), suggesting that the Get SET Early approach detects females with ASD at expected, or better than expected, rates. Other possible issues relating to the use of the Get SET Early model is the fact that it is unlikely that extremely-high-functioning individuals with ASD will be detected using developmental screening tools, such as the CSBS, that rely on the presence of observable symptoms. Indeed, although precise estimates regarding late-identified cases of ASD are not formally established, some studies suggest that as many as 6–25% of ASD individuals do not receive a diagnosis until school age or later[59,60]. It is thus possible that results from the present study may, or may not, generalize to the subset of individuals with ASD who have mild presentations and do not receive diagnoses until later in life[60–62]. Future studies can examine this possibility by examining sex differences in late-detected groups compared with individuals from more traditionally detected cohorts.

Collectively, despite limitations inherent in past studies of sex differences, such as relatively small sample sizes (for example, n = 28–96), limited clinical data and a lack of longitudinal data, these studies nonetheless found few to no sex differences in important clinical measures in ASD at early ages. The present study overcomes these limitations and also did not find compelling evidence of sex differences in the clinical presentation and progression of ASD during the earliest years of the disorder. ASD is highly heritable[56], yet more than 80–90% of patients are classified as idiopathic with no identifiable genetic cause. Thus, whether or not there might be genetic differences between girls and boys with idiopathic ASD is currently unknown, but if there are differences, they do not result in clinical sex differences at early ages. Future studies aimed at identifying whether sex differences emerge at later ages in autism should incorporate comprehensive and reproducible designs, such as those used herein. In conclusion, although later environmental influences could arguably impact later-age symptom presentation, particularly in females[32,63,64], the current body of evidence and our present study do not support previous speculations about sex differences in ASD at the time of first diagnosis. Therefore, it is unlikely that girls with ASD differ clinically from boys with ASD across early development.

## Methods

### Study design

Sex differences were examined using both cross-sectional and longitudinal clinical data collected between 2002 and 2022. Participants in the current study completed comprehensive psychological assessments on social skills, expressive and receptive language, gesture production, motor skills, visual reception and core ASD symptoms ('Clinical testing' section). This research met all of the ethical requirements of the Human Research Protection Program under the approval of the University of California, San Diego Office of IRB Administration (project number 202115). Parents gave written informed consent and all testing occurred at the University of California, San Diego Autism Center of Excellence.

### Participants, recruitment, clinical testing and diagnostic criteria

**Participants.** A total of 2,618 toddlers participated in this study, including those with a diagnosis of ASD (n = 1,539; 1,200 male and 339 female; mean age = 28.6 months), those with DD (n = 478; 349 male and 129 female; mean age = 26.0 months) and TD individuals (n = 601; 349 male and 252 female; mean age = 25.7 months).

No statistical methods were used to predetermine sample sizes, but our sample sizes are larger than those reported in previous publications[4,14,65].

Within this cohort, 44.3% returned for one or more clinical test session before the age of 4 years, resulting in data from a total of 4,440 longitudinal testing sessions. Socioeconomic status (that is, median household income) and racial and ethnic distributions were as expected for the San Diego region (Table 1). Overall, no significant sex differences in socioeconomic status were found; however, when examining children with ASD and TD individuals separately, boys had a higher median household income, whereas no sex differences were observed among children with DD (Supplementary Table 14).

**Recruitment.** The majority of toddlers (~75%) were recruited through a general population-based screening approach using the Get SET Early method (formerly known as the 1-Year Well-Baby Check-Up Approach[66]. This method results in the detection of ASD in children as young as 12 months. The programme is based on the collaboration of more than 200 local paediatricians who screen for ASD and other delays using the CSBS Infant-Toddler Checklist at all 12-, 18- and 24-month well-baby check-ups and refer toddlers to our centre who fail the CSBS screening and/or are suspected of having ASD. The remainder of the cohort were community referrals who contacted our centre seeking a developmental evaluation. Toddlers referred younger than 30 months were invited for a re-evaluation every 9–12 months until their third birthday, when a final diagnosis was given. Toddlers were stratified into the diagnostic groups described above based on the results from their most recent diagnostic evaluation.

**Clinical testing.** Toddlers and their parents participated in a series of tests, including the ADOS-2 (ref. 42) or ADOS-G, MSEL[67], Vineland-3 (ref. 41), MacArthur-Bates CDI[43], CSBS[39] and GeoPref eye-tracking test. The study examined three subscales of the ADOS: social affect, RRB and overall total. Additionally, five subscales of the Vineland were assessed: communication, daily living skills, socialization, motor skills and ABC. For the MSEL, the following five subscales were evaluated: visual reception, fine motor skills, expressive language, receptive language and ELC. The CDI–WG included five subscales: words produced, words understood, early gestures, later gestures and total gestures. Words produced from the CDI–WS were also assessed. Furthermore, the CSBS assessment covered four subscales: social composite, speech composite, symbolic composite and total score. Lastly, the percentage of social fixation in the GeoPref test was also examined. Approximately 80% of the GeoPref eye-tracking data came from our earlier eye-tracking study[40] and the remaining 20% were collected subsequently.

All assessments were administered by licensed clinical psychologists and eye-tracking technicians blind to the initial CSBS screening scores. To help ensure interclinician reliability, the lead clinical psychologist (C.C.B.; an ADOS-certified independent trainer with over 25 years of experience in toddler testing) was responsible for training the other psychologists to achieve research-reliable levels on the ADOS. ADOS reliability checks were conducted approximately twice per year. The consistency of testing procedures and setting and the use of only licensed clinical psychologists may have served to bolster the validity of the results (see Supplementary Information for more details).

**Diagnostic criteria.** A toddler was assigned to one of the following diagnostic categories based on the following criteria: ASD (scored within the range of concern on the ADOS-2 and was considered to have ASD based on *Diagnostic and Statistical Manual of Mental Disorders* (5th edn) criteria and clinical judgement); DD (scored ≤85 on the overall MSEL ELC); or TD individuals (scored within the normal range on all clinical assessments).

## Data analyses

Three main analyses were conducted, including a primary cross-sectional analysis, a longitudinal analysis and a cluster analysis. In all subsequent sections, the statistical tests used were two sided.

**Primary analysis.** The primary analysis included examination of sex differences between the group with ASD and the TD group across all tests and subscales defined in the Methods. Although the primary goal of the study was to examine possible sex differences in children with ASD specifically, examination of possible sex differences within TD toddlers was also included to aid interpretation of the results. Examination of all available data revealed a statistically significant ($P < 0.001$) 4-month age difference between toddlers with ASD (mean age = 29.4 months) and TD individuals (mean age = 25.2 months), except for the CDI–WS test. To ensure that any reported differences in ASD and TD children were not driven by age effects, primary analyses included data from all available time points, ranging from visit 1 to visit 5, representing the number of times each child visited the clinic. Cardinality matching[68] was then conducted to achieve an evenly age-matched sample. Cardinality matching is an alternative to propensity score matching that resolves the covariate overlap problem. It identifies the largest possible matched sample based on the pre-specified ratio of participants with ASD to TD toddlers and balance criteria, such as age[69]. The MatchIt package[70] in R was used to perform cardinality matching. Matching was conducted on different tests separately, as some of the participants took the tests at slightly different ages and there was also small variation in the sample sizes of groups with ASD and TD toddlers across tests (Table 1). The best ratio of participants with ASD to TD toddlers was found to be 2.0 for ADOS and Vineland, 1.5 for MSEL and 1.0 for CDI–WG. No matching was required for CDI–WS since there were no significant age differences between the group with ASD and the TD group.

First, we conducted a two-way ANOVA including sex, group and sex × group interaction across all tests and subscales to explore the relationship between groups and sex on clinical tests. If there was a significant interaction or a significant sex main effect, we proceeded to test our planned contrasts. Since we were interested in sex differences in the groups with ASD and TD individuals, we focused on two out of six multiple comparisons resulting from the ANOVA interaction (that is, ASD (F) versus ASD (M) and TD (F) versus TD (M)) and adjusted the $P$ values for two comparisons using the false discovery rate[71]. Then, depending on the data normality and homogeneity of variance assumptions, examination of sex differences within the group with ASD and the TD group was conducted using either $t$-tests or Kruskal–Wallis rank-sum tests as appropriate. Effect sizes for Kruskal–Wallis rank-sum tests are reported as $\eta^2$ values, where 0.01–0.05 represents a small effect, 0.06–0.13 represents a medium effect and >0.14 represents a large effect based on the recommendation of Lomax and Hahs-Vaughn[72]. Effect sizes for $t$-tests are reported as Cohen's $d$ values, where small, medium and large effects correspond to approximate ranges of 0.2–0.49, 0.5–0.79 and ≥0.8, respectively[73]. In a separate analysis, toddlers with DD who had a MSEL ELC score of ≤85 were examined for potential sex differences.

For the primary analysis, we examined sex differences in groups with ASD, TD and DD across multiple subscales using the appropriate statistical tests. When the assumptions of normality and homogeneity of variance were satisfied, two-sample $t$-tests were employed to compare group means. CIs for the mean differences were calculated based on the standard error and the critical values from the $t$-distribution at a 95% confidence level. In cases where the assumptions of the $t$-test were violated, non-parametric Kruskal–Wallis tests were used to assess group differences. Here CIs were calculated to reflect the rank-based differences between groups, providing a range of plausible differences between the medians of the groups.

**Cluster analysis.** To better understand patterns of clinical heterogeneity and whether or not they may be different across males and females

with ASD, we conducted a study female and male clinical subtypes in ASD and TD using SNF[44]. Since the SNF method requires complete data across all layers and cannot accommodate missing information, we included only those participants who were consistently present across the ADOS, Vineland and MSEL subscales after matching. This process resulted in a consolidated age-matched sample of 1,673 participants with ASD or TD toddlers.

We employed Pearson correlation—a filter method in feature selection techniques—to identify features for inclusion in the SNF analysis. We calculated all possible pairwise correlations among the subscales of ADOS, Vineland and MSEL and then removed those subscales that were highly correlated with others. Additionally, we grouped similar subscales that approximately measure the same concept into a separate layer. Therefore, data from three clinical domains, including social, language and motor, were considered as three layers in SNF. Within the social domain, we considered the ADOS social affect as well as the Vineland socialization subscales. For the language domain, we included the MSEL receptive language, MSEL expressive language and Vineland communication subscales. Lastly, the motor domain included Vineland motor skills and MSEL fine motor skills. The data were normalized at each layer, and given that the data are continuous the Euclidean distance was utilized to compute pairwise distances. Upon fusing the similarity graphs through SNF, spectral clustering[74] (a community detection algorithm) was employed to identify cluster labels. To determine the optimal number of clusters (that is, three) based on various distance measures and clustering methods, we compared the results of 24 indices using the NbClust[75] R package, the Bayesian information criterion based on a Gaussian mixture model in the mclust[76] R package and the total within-cluster sums of squares (that is, elbow method) using the cluster[77] R package (Supplementary Table 2). In addition to the primary SNF analysis, we conducted a separate SNF analysis incorporating sex as a main feature. This allowed us to further examine the effect of sex on the clustering outcomes.

Training SNF was performed on 80% of the data ($n = 1,337$) and the remaining 20% ($n = 336$) were held out to test the clusters obtained. To assess the quality of the clusters during both training and testing stages, we computed silhouette scores[78], which ranged from −1 to 1. Higher scores indicate better-defined and more well-separated clusters. The resulting scores of 0.46 for training and 0.4 for testing suggest reasonably well-separated clusters in both stages (Fig. 4c). Furthermore, to investigate the differences between clusters, we performed ANOVA and multiple pairwise comparisons with false discovery rate correction on several variables included in the SNF, as well as other variables outside of the SNF, to externally validate the clusters. To evaluate the robustness of SNF clustering, we randomly removed 5, 10, 20, 30, 40 and 50% of the data, conducted SNF on the remaining data and calculated NMI values. We repeated this process 100 times for each random removal proportion to obtain more stable results. Finally, a five-fold cross-validation was conducted on the training set, repeated ten times, to further validate the clusters.

After validating the obtained clusters using the aforementioned strategies, we took an additional step to investigate the presence of sex differences at the subtype level. This involved examining variables within the training and test data, as well as external variables, using the $t$-test or the Kruskal–Wallis test as appropriate. Furthermore, we explored the associations between sex and cluster membership in both the group with ASD and the TD group separately, employing the chi-squared test for this purpose.

**Longitudinal analyses using a latent growth model with individually varying times of observation.** We began by investigating sex differences through an initial cross-sectional analysis. Subsequently, we evaluated potential sex differences within ASD and TD subtypes. Finally, to comprehensively analyse sex differences in toddlers with ASD and TD individuals, we conducted a longitudinal analysis to examine these differences over time. Given the availability of longitudinal

scores, ADOS, Vineland and MSEL subscales were used in this analysis. However, due to inconsistencies between test intervals, which could potentially lead to overestimation or underestimation of the targeted parameters[79], we utilized latent growth modelling[80]. This method was chosen for its ability to accommodate individual variations in observation times.

In this multilevel model, repeated measures were nested within participants, and sex was a predictor of the random intercept and random slope at level two. Level one variables were subscale scores from the ADOS, Vineland and MSEL, and Cohen's *d* was reported as the effect size[73]. Maximum likelihood with robust standard errors was the estimation method, but due to a convergence problem in three subscales (that is, Vineland daily living skills, Vineland ABC and MSEL visual reception) in the group with ASD, the MLF estimator was used (that is, a simpler version of maximum likelihood with robust standard errors) for the calculation of standard errors. MLF estimator settings represent "maximum likelihood parameter estimates with standard errors approximated by first-order derivatives and a conventional chi-square test statistic"[81]. Additionally, for some subscales, such as ADOS RRB and all four subscales of MSEL for the TD group, only up to four measurements were considered due to low sample sizes. Specifically, the fifth measurement had a variance of zero, which made it unusable in the analysis. CIs for the parameter estimates, including intercepts and slopes, were computed in Mplus using the standard errors derived from the maximum likelihood estimation. These CIs represent the range of plausible values for the growth parameters at the specified confidence level (95%). Data preparation was conducted using R and the analyses were performed using Mplus version 8.3 (ref. 81).

### Reporting summary

Further information on research design is available in the Nature Portfolio Reporting Summary linked to this article.

## Data availability

The data supporting the findings of this study are available on the Autism Center of Excellence laboratory's GitHub page at https://github.com/ACE-UCSD/Autism-Sex-Differences-Analysis-Pathway. Source data are provided with this paper.

## Code availability

The codes supporting the findings of this study are available on the Autism Center of Excellence laboratory's GitHub page at https://github.com/ACE-UCSD/Autism-Sex-Differences-Analysis-Pathway.

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

## Acknowledgements

The present study was supported by the National Institutes of Health (award numbers R01MH118879, R01MH080134, R01MH10446 and R01MH121595 to K.P. and P50-MH081755, R01MH110558 and R01DC016385 to E.C.). The content presented in the present study is solely the responsibility of the authors and does not necessarily represent the official views of the National Institutes of Health. The funders had no role in study design, data collection and analysis, decision to publish or preparation of the manuscript.

## Author contributions

K.P. and E.C. conceived of the idea, designed the study and analysis concepts and obtained funding. S. Nazari was responsible for conceiving of, performing and interpreting the statistical analyses, with input from K.P., E.C., J.Z. and M.V.L. S.R.C. wrote an initial draft of the manuscript. S. Nazari., K.P. and E.C. wrote the final manuscript. All authors contributed to the revisions and editing of the manuscript and read and approved the final version. S.R.C., S. Nalabolu, C.C.B., C.A., A.E., S.J.A., A.G. and L.L. were instrumental in collecting the data, recruiting participants and administering the tests.

## Competing interests

The authors declare no competing interests.

## Additional information

**Extended data** is available for this paper at https://doi.org/10.1038/s41562-025-02132-6.

**Correspondence and requests for materials** should be addressed to Eric Courchesne or Karen Pierce.

¹Department of Neurosciences and Autism Center of Excellence, University of California, San Diego, La Jolla, CA, USA. ²Laboratory for Autism and Neurodevelopmental Disorders, Center for Neuroscience and Cognitive Systems, Istituto Italiano di Tecnologia, Rovereto, Italy. ³These authors jointly supervised this work: Eric Courchesne, Karen Pierce. ✉e-mail: ecourchesne@health.ucsd.edu; kpierce@health.ucsd.edu

**Extended Data Table 1 | Primary analysis of sex differences in ASD and TD toddlers**

| Test | Subscale | Statistic[a] | P-value | Adj P-value | Effect size | Difference | CI-lower | CI-upper |
|------|----------|-----------|---------|-------------|-------------|------------|----------|----------|
| ADOS | ASD-Social Affect | 3.72 | 0.054 | 0.054 | - | 1.00 | 0.00 | 1.00 |
| | TD-Social Affect | 4.41 | 0.036 | 0.054 | - | 0.00 | -1.00 | 0.00 |
| | ASD-R & R Behavior | 1.53 | 0.216 | 0.216 | - | 0.00 | 0.00 | 0.00 |
| | TD-R & R Behavior | 3.58 | 0.058 | 0.117 | - | 0.00 | 0.00 | 0.00 |
| | ASD-Overall Total | 1.13 | 0.287 | 0.287 | - | 0.00 | 0.00 | 1.00 |
| | TD-Overall Total | 5.65 | 0.017 | 0.035* | 0.01 | 0.00 | -1.00 | 0.00 |
| | | | | | | | | |
| Vineland | ASD-Communication | 0.55 | 0.458 | 0.458 | - | 1.00 | -1.00 | 3.00 |
| | TD-Communication | 4.74 | 0.029 | 0.059 | - | 2.00 | 0.00 | 3.00 |
| | ASD-Daily Living Skills | 5.43 | 0.020 | 0.020* | < 0.01 | 2.00 | 0.00 | 4.00 |
| | TD-Daily Living Skills | 17.97 | < .001 | < .001* | 0.03 | 4.00 | 2.00 | 6.00 |
| | ASD-Motor Skills | 0.01 | 0.930 | 0.930 | - | 0.00 | -2.00 | 1.00 |
| | TD-Motor Skills | 0.06 | 0.803 | 0.930 | - | 0.00 | -1.00 | 2.00 |
| | ASD-Socialization | 1.73 | 0.188 | 0.188 | - | 1.00 | 0.00 | 3.00 |
| | TD-Socialization | 7.02 | 0.008 | 0.016* | 0.01 | 2.00 | 0.00 | 4.00 |
| | ASD-ABC | 1.31 | 0.253 | 0.253 | - | 1.00 | -1.00 | 2.00 |
| | TD-ABC | 9.93 | 0.002 | 0.003* | 0.02 | 3.00 | 1.00 | 4.00 |
| | | | | | | | | |
| MSEL | ASD-Fine Motor | 2.87 | 0.090 | 0.090 | - | 2.24 | -0.37 | 4.88 |
| | TD-Fine Motor | 10.98 | 0.001 | 0.002* | 0.02 | 4.03 | 1.66 | 6.41 |
| | ASD-Visual Reception | 1.26 | 0.262 | 0.262 | - | -1.77 | -4.88 | 1.33 |
| | TD-Visual Reception | 6.36 | 0.012 | 0.023* | 0.01 | 3.76 | 0.80 | 6.66 |
| | ASD-Receptive Language | 0.95 | 0.331 | 0.331 | - | -2.35 | -7.46 | 2.41 |
| | TD-Receptive Language | 3.02[b] | 0.003 | 0.005* | 0.26 | 4.80 | 1.67 | 7.93 |
| | ASD-Expressive Language | 1.16 | 0.282 | 0.282 | - | 2.16 | -1.76 | 6.18 |
| | TD-Expressive Language | 2.66 | 0.103 | 0.206 | - | 2.25 | -0.48 | 5.10 |
| | | | | | | | | |
| CDI-WG | ASD-Words Produced | 3.98 | 0.046 | 0.092 | - | 0.00 | 0.00 | 1.00 |
| | TD-Words Produced | 0.56 | 0.455 | 0.455 | - | 1.00 | -1.00 | 3.00 |
| | ASD-Words Understood | 3.85 | 0.050 | 0.100 | - | 7.00 | 0.00 | 15.00 |
| | TD-Words Understood | 2.28 | 0.131 | 0.131 | - | 12.00 | -4.00 | 28.00 |
| | ASD-Early Gestures | 0.79[b] | 0.433 | 0.433 | - | 0.43 | -0.66 | 1.52 |
| | TD-Early Gestures | 4.77 | 0.029 | 0.058 | - | 1.00 | 0.00 | 2.00 |
| | ASD-Later Gestures | 0.60 | 0.440 | 0.440 | - | 1.00 | -1.00 | 3.00 |
| | TD-Later Gestures | 5.12 | 0.024 | 0.047* | 0.02 | 3.00 | 0.00 | 6.00 |
| | ASD-Total Gestures | 0.78 | 0.377 | 0.377 | - | 1.00 | -2.00 | 4.00 |
| | TD-Total Gestures | 2.38[b] | 0.018 | 0.036* | 0.32 | 4.03 | 0.70 | 7.37 |
| | | | | | | | | |
| CDI-WS | ASD-Words Produced | 0.43 | 0.514 | 0.514 | - | -1.00 | -3.00 | 2.00 |
| | TD-Words Produced | 5.22 | 0.022 | 0.045* | 0.02 | 55.00 | 7.00 | 108.00 |
| | | | | | | | | |
| CSBS | ASD-Social Composite | 5.00 | 0.025 | 0.025* | 0.01 | 1.00 | 0.00 | 7.00 |
| | TD-Social Composite | 5.67 | 0.017 | 0.025* | 0.01 | 7.00 | 0.00 | 13.00 |
| | ASD-Speech Composite | 3.60 | 0.058 | 0.058 | - | 1.00 | 0.00 | 7.00 |
| | TD- Speech Composite | 4.80 | 0.028 | 0.057 | - | 4.00 | 0.00 | 12.00 |
| | ASD-Symbolic Composite | 8.44 | 0.004 | 0.007* | 0.02 | 4.00 | 0.00 | 4.00 |
| | TD-Symbolic Composite | 3.93 | 0.047 | 0.047* | 0.01 | 4.00 | 0.00 | 9.00 |
| | ASD-Total Score | 7.32 | 0.007 | 0.014* | 0.01 | 2.00 | 0.00 | 5.00 |
| | TD- Total Score | 5.18 | 0.023 | 0.023* | 0.01 | 5.00 | 0.00 | 10.00 |
| | | | | | | | | |
| GeoPref | ASD-Social % Fixation | 0.02 | 0.886 | 0.886 | - | 0.42 | -4.77 | 5.71 |
| | TD-Social % Fixation | 0.31 | 0.579 | 0.886 | - | 0.89 | -2.11 | 4.00 |

*Note.* [a] = Statistic is Kruskal-Wallis chi-squared test and its reported effect size is Eta squared. [b] = Statistic is t-test and its reported effect size is Cohen's *d*. \* = *P* < .05. *Adj P-value* = Multiple comparisons were corrected by FDR. All the tests are two-sided. *Difference* = Difference in location or means. R & R Behavior = Restricted & Repetitive Behavior. ABC = Adaptive Behavior Composite. CI: 95% Confidence Interval.

**Extended Data Table 2 | Primary analysis of sex differences in DD toddlers**

| Test | Subscale | Statistic[a] | P-value | Effect size | Difference | CI-lower | CI-upper |
|------|----------|-----------|---------|-------------|------------|----------|----------|
| ADOS | Social Affect | 2.08 | 0.150 | - | 0.00 | 0.00 | 1.00 |
| | R & R Behavior | 5.22 | 0.022* | < 0.01 | 0.00 | -1.00 | 0.00 |
| | Overall Total | 0.21 | 0.645 | - | 0.00 | -1.00 | 1.00 |
| | | | | | | | |
| Vineland | Communication | 0.01 | 0.942 | - | 0.00 | -3.00 | 3.00 |
| | Daily Living Skills | 0.36 | 0.547 | - | -1.00 | -3.00 | 2.00 |
| | Motor Skills | 4.07 | 0.044* | 0.01 | -3.00 | -5.00 | 0.00 |
| | Socialization | 0.00 | 0.973 | - | 0.00 | -2.00 | 2.00 |
| | Adaptive Behavior Composite | 0.35 | 0.556 | - | -1.00 | -3.00 | 2.00 |
| | | | | | | | |
| MSEL | Fine Motor | 0.74 | 0.390 | - | -1.31 | -4.40 | 1.77 |
| | Visual Reception | 0.71 | 0.400 | - | -1.25 | -4.35 | 1.81 |
| | Receptive Language | 0.75 | 0.386 | - | 1.33 | -1.85 | 4.60 |
| | Expressive Language | 1.90 | 0.168 | - | 2.95 | -1.21 | 7.16 |
| | | | | | | | |
| CDI-WG | Words Produced | 2.03 | 0.154 | - | -1.00 | -2.00 | 0.00 |
| | Words Understood | 3.36 | 0.067 | | -14.00 | -34.00 | 2.00 |
| | Early Gestures | 1.22 | 0.269 | - | -1.00 | -3.00 | 1.00 |
| | Later Gestures | 1.34 | 0.246 | - | -2.00 | -6.00 | 2.00 |
| | Total Gestures | -0.91[b] | 0.371 | - | -2.68 | -8.71 | 3.35 |
| | | | | | | | |
| CDI-WS | ASD-Words Produced | 2.08 | 0.149 | - | 12.00 | -6.00 | 25.00 |
| | | | | | | | |
| CSBS | Social Composite Score | 0.17 | 0.683 | - | 0.00 | -7.00 | 3.00 |
| | Speech Composite Score | 0.25 | 0.618 | - | 0.00 | -4.00 | 1.00 |
| | Symbolic Composite Score | 0.00 | 0.946 | - | 0.00 | -4.00 | 4.00 |
| | Total Score | 0.07 | 0.785 | - | 0.00 | -3.00 | 2.00 |
| | | | | | | | |
| GeoPref | ASD-Social % Fixation | 0.02 | 0.883 | - | 0.54 | -5.03 | 6.14 |

*Note.* [a] = Statistic is Kruskal-Wallis chi-squared test and its reported effect size is Eta squared. [b] = Statistic is t-test and its reported effect size is Cohen's $d$, * = $P < .05$. All the tests are two-sided. *Difference* = Difference in location or means. CI: 95% Confidence Interval. R & R Behavior = Restricted & Repetitive Behavior.

**Extended Data Table 3 | SNF train set with sex differences**

| **ASD** | | | | Social domain | | Motor domain | | Language domain | | | | |
|---|---|---|---|---|---|---|---|---|---|---|---|---|
| Cluster | Diagnosis | Sex | Age | SA | SOC | MTR | FM | COM | RL | EL | N | |
| All data | ASD | F | 25.3 | 14.6 | 83.4 | 90.4 | 81.8 | 75.3 | 49.5 | 53.9 | 206 | |
| | ASD | M | 25.8 | 13.7 | 82.3 | 91.4 | 80.3 | 74.0 | 54.1 | 53.8 | 686 | % |
| High (C1) | ASD | F | 20.7 | 9.7 | 95.7 | 100.3 | 109.8 | 95.9 | 88.2 | 83.2 | 21 | 10.2% |
| | ASD | M | 20.1 | 8.2 | 99.4 | 101.7 | 106.0 | 95.4 | 95.4 | 81.3 | 67 | 9.8% |
| Medium (C2) | ASD | F | 25.4 | 14.7 | 85.4 | 91.8 | 83.1 | 77.1 | 50.7 | 54.7 | 147 | 71.4% |
| | ASD | M | 25.8 | 13.6 | 83.1 | 92.3 | 81.5 | 75.6 | 55.2 | 54.9 | 515 | 75.1% |
| Low (C3) | ASD | F | 27.6 | 17.2 | 68.9 | 79.1 | 61.3 | 56.7 | 23.7 | 34.9 | 38 | 18.4% |
| | ASD | M | 29.6 | 17.8 | 67.3 | 80.4 | 58.0 | 52.7 | 21.9 | 30.2 | 104 | 15.2% |
| **TD** | | | | Social domain | | Moto domain | | Language domain | | | | |
| Cluster | Diagnosis | Sex | Age | SA | SOC | MTR | FM | COM | RL | EL | N | |
| All data | TD | F | 25.5 | 2.2 | 105.6 | 101.3 | 107.2 | 103.9 | 110.3 | 103.3 | 186 | |
| | TD | M | 27.0 | 2.5 | 102.9 | 99.7 | 101.8 | 102.3 | 104.5 | 101.7 | 259 | % |
| High (C1) | TD | F | 25.5 | 2.1 | 106.2 | 102.0 | 108.0 | 104.8 | 111.5 | 104.7 | 179 | 96.2% |
| | TD | M | 27.0 | 2.3 | 103.4 | 100.2 | 102.2 | 102.8 | 105.2 | 102.3 | 248 | 95.8% |
| Medium (C2) | TD | F | 24.0 | 4.0 | 95.6 | 87.0 | 90.3 | 85.6 | 85.5 | 74.0 | 7 | 3.8% |
| | TD | M | 26.6 | 6.6 | 91.9 | 89.1 | 91.4 | 90.6 | 87.3 | 87.6 | 11 | 4.2% |

*Note.* SA: ADOS Social Affect, SOC: Vineland Socialization, MTR: Vineland Motor Skills, FM: Std. MSEL Fine Motor, COM: Vineland Communication, RL: Std. MSEL Receptive Language, EL: Std. MSEL Expressive Language. Yellow highlighted cells show sex differences with darker color indicating higher means.

**Extended Data Table 4 | SNF test set with sex differences**

| **ASD** | | | | Social domain | | Motor domain | | Language domain | | | | |
|---|---|---|---|---|---|---|---|---|---|---|---|---|
| Cluster | Diagnosis | Sex | Age | SA | SOC | MTR | FM | COM | RL | EL | N | |
| All data | ASD | F | 25.3 | 14.2 | 81.0 | 90.5 | 84.7 | 76.0 | 59.6 | 58.3 | 48 | |
| | ASD | M | 26.0 | 13.6 | 82.0 | 90.3 | 79.5 | 74.5 | 53.6 | 52.7 | 176 | % |
| High (C1) | ASD | F | 28.5 | 8.3 | 96.3 | 99.7 | 112.1 | 104.0 | 116.6 | 97.4 | 3 | 6.3% |
| | ASD | M | 20.0 | 5.3 | 96.7 | 97.3 | 97.4 | 94.2 | 89.9 | 84.4 | 16 | 9.1% |
| Medium (C2) | ASD | F | 24.9 | 14.5 | 80.2 | 90.4 | 84.3 | 74.8 | 57.0 | 56.7 | 44 | 91.7% |
| | ASD | M | 26.4 | 14.1 | 81.7 | 90.5 | 79.3 | 74.6 | 52.6 | 51.2 | 150 | 85.2% |
| Low (C3) | ASD | F | 34.1 | 19.0 | 70.0 | 67.0 | 17.5 | 42.0 | 2.9 | 11.7 | 1 | 2.1% |
| | ASD | M | 29.4 | 19.4 | 63.4 | 76.5 | 54.5 | 41.4 | 10.8 | 25.4 | 10 | 5.7% |
| **TD** | | | | Social domain | | Moto domain | | Language domain | | | | |
| Cluster | Diagnosis | Sex | Age | SA | SOC | MTR | FM | COM | RL | EL | N | |
| All data | TD | F | 25.2 | 2.6 | 105.8 | 99.7 | 104.8 | 102.3 | 106.5 | 103.0 | 48 | |
| | TD | M | 25.9 | 3.3 | 104.3 | 99.3 | 102.1 | 103.1 | 102.2 | 100.5 | 64 | % |
| High (C1) | TD | F | 25.1 | 2.5 | 106.2 | 100.8 | 105.1 | 102.9 | 107.6 | 103.4 | 46 | 95.8% |
| | TD | M | 26.9 | 3.0 | 105.4 | 100.8 | 103.8 | 104.9 | 104.8 | 101.0 | 57 | 89.1% |
| Medium (C2) | TD | F | 26.7 | 5.0 | 97.0 | 75.5 | 98.1 | 87.5 | 80.3 | 93.0 | 2 | 4.2% |
| | TD | M | 17.6 | 5.9 | 94.9 | 86.7 | 88.3 | 88.1 | 81.1 | 95.9 | 7 | 10.9% |

*Note.* SA: ADOS Social Affect, SOC: Vineland Socialization, MTR: Vineland Motor Skills, FM: Std. MSEL Fine Motor, COM: Vineland Communication, RL: Std. MSEL Receptive Language, EL: Std. MSEL Expressive Language. Yellow highlighted cells show sex differences with darker color indicating higher means.

**Extended Data Table 5 | SNF external validation with sex differences**

| ASD Cluster | Diagnosis | Sex | External Variables RR | W.Prod | W.Und | Early.Ges | Later.Ges | Tot.Ges | WS.W.Prod | % Social Fixation |
|---|---|---|---|---|---|---|---|---|---|---|
| All data | ASD | F | 4.8 | 62.6 | 55.5 | 8.1 | 10.6 | 18.7 | 62.6 | 56.80 |
|  | ASD | M | 4.9 | 64.0 | 39.0 | 7.5 | 9.8 | 17.3 | 64.0 | 56.29 |
| High | ASD | F | 3.3 | 192.8 | 59.0 | 9.2 | 10.7 | 19.9 | 192.8 | 70.77 |
| (C1) | ASD | M | 3.1 | 169.7 | 61.7 | 9.6 | 13.5 | 23.0 | 169.7 | 72.81 |
| Medium | ASD | F | 4.8 | 64.3 | 52.7 | 7.3 | 10.7 | 18.0 | 64.3 | 57.26 |
| (C2) | ASD | M | 4.8 | 53.5 | 30.3 | 6.8 | 8.5 | 15.2 | 53.5 | 56.55 |
| Low | ASD | F | 5.7 | 2.9 | 80.0 | 12.0 | 7.0 | 19.0 | 2.9 | 43.17 |
| (C3) | ASD | M | 6.4 | 18.8 | 19.5 | 4.5 | 5.8 | 10.3 | 18.8 | 39.89 |

| TD Cluster | Diagnosis | Sex | External Variables RR | W.Prod | W.Und | Early.Ges | Later.Ges | Tot.Ges | WS.W.Prod | % Social Fixation |
|---|---|---|---|---|---|---|---|---|---|---|
| All data | TD | F | 0.5 | 293.8 | 84.6 | 10.9 | 16.2 | 27.1 | 293.8 | 75.39 |
|  | TD | M | 0.7 | 238.0 | 72.0 | 10.1 | 13.9 | 24.0 | 238.0 | 74.47 |
| High | TD | F | 0.5 | 299.8 | 87.1 | 11.0 | 16.4 | 27.4 | 299.8 | 75.25 |
| (C1) | TD | M | 0.6 | 240.8 | 72.3 | 10.2 | 14.0 | 24.1 | 240.8 | 74.54 |
| Medium | TD | F | 0.1 | 36.0 | 23.8 | 8.5 | 13.3 | 21.8 | 36.0 | 80.24 |
| (C2) | TD | M | 1.6 | 168.2 | 60.7 | 8.3 | 11.0 | 19.3 | 168.2 | 72.77 |

*Note.* RR: ADOS Restricted and Repetitive Behavior, W.Prod: CDI-WG-Words Produced, W.Und: CDI-WG-Words Understood, Early.Ges: CDI-WG-Early Gestures, Later.Ges: CDI-WG-Later Gestures, Tot.Ges: CDI-WG-Total Gestures, WS.W.Prod: CDI-WS-Words Produced, % Social Fixation: GeoPref Eye Tracking Test. Yellow highlighted cells show sex differences with darker color indicating higher means.

**Extended Data Table 6 | Longitudinal analysis result**

| Test | Differences in male and female groups | Parameter | Estimate | CI-lower | CI-upper | P-value | Cohen's d |
|---|---|---|---|---|---|---|---|
| ADOS | ASD-Social Affect | Intercept | 0.457 | -0.027 | 0.941 | 0.120 | - |
| | | Slope | -0.05 | -0.09 | -0.01 | 0.047* | 0.42 |
| | TD-Social Affect | Intercept | -0.73 | -1.03 | -0.42 | < .001* | 0.59 |
| | | Slope | 0.03 | 0.01 | 0.05 | 0.026* | 0.79 |
| | ASD-R & R Behavior | Intercept | -0.107 | -0.324 | 0.110 | 0.418 | - |
| | | Slope | -0.013 | -0.036 | 0.009 | 0.321 | - |
| | TD-R & R Behavior | Intercept | -0.25 | -0.37 | -0.13 | 0.001* | 0.48 |
| | | Slope | 0.007 | -0.002 | 0.017 | 0.207 | - |
| | ASD-Overall Total | Intercept | 0.392 | -0.206 | 0.990 | 0.280 | - |
| | | Slope | -0.07 | -0.12 | -0.02 | 0.020* | 0.44 |
| | TD-Overall Total | Intercept | -0.95 | -1.30 | -0.60 | < .001* | 0.68 |
| | | Slope | 0.03 | 0.01 | 0.06 | 0.020* | 0.67 |
| | | | | | | | |
| Vineland | ASD-Communication | Intercept | 0.745 | -0.764 | 2.254 | 0.417 | - |
| | | Slope | -0.090 | -0.261 | 0.081 | 0.386 | - |
| | TD-Communication | Intercept | 2.76 | 1.33 | 4.20 | 0.002* | 0.35 |
| | | Slope | -0.078 | -0.192 | 0.037 | 0.266 | - |
| | ASD-Daily Living Skills | Intercept | 1.636 | 0.258 | 3.015 | 0.051 | - |
| | | Slope | -0.079 | -0.213 | 0.056 | 0.337 | - |
| | TD-Daily Living Skills | Intercept | 3.78 | 2.32 | 5.24 | < .001* | 0.49 |
| | | Slope | 0.083 | -0.032 | 0.197 | 0.234 | - |
| | ASD-Motor Skills | Intercept | -0.826 | -1.934 | 0.281 | 0.220 | - |
| | | Slope | 0.14 | 0.03 | 0.26 | 0.046* | 0.43 |
| | TD-Motor Skills | Intercept | 0.006 | -1.233 | 1.244 | 0.994 | - |
| | | Slope | 0.16 | 0.06 | 0.26 | 0.008* | 0.55 |
| | ASD-Socialization | Intercept | 0.120 | -1.113 | 1.353 | 0.873 | - |
| | | Slope | 0.134 | 0.004 | 0.265 | 0.091 | - |
| | TD-Socialization | Intercept | 2.07 | 0.86 | 3.28 | 0.005* | 0.30 |
| | | Slope | 0.045 | -0.067 | 0.157 | 0.512 | - |
| | ASD-ABC | Intercept | 0.263 | -0.912 | 1.439 | 0.712 | - |
| | | Slope | 0.021 | -0.096 | 0.137 | 0.769 | - |
| | TD-ABC | Intercept | 2.46 | 1.20 | 3.73 | 0.001* | 0.36 |
| | | Slope | 0.054 | -0.053 | 0.161 | 0.410 | - |
| | | | | | | | |
| MSEL | ASD-Fine Motor | Intercept | 1.387 | -0.487 | 3.262 | 0.224 | - |
| | | Slope | -0.068 | -0.223 | 0.087 | 0.468 | - |
| | TD-Fine Motor | Intercept | 1.830 | -0.453 | 4.113 | 0.187 | - |
| | | Slope | 0.162 | -0.017 | 0.341 | 0.136 | - |
| | ASD-Visual Reception | Intercept | -2.174 | -4.467 | 0.119 | 0.119 | - |
| | | Slope | 0.152 | -0.082 | 0.385 | 0.285 | - |
| | TD-Visual Reception | Intercept | 4.12 | 1.84 | 6.40 | 0.003* | 0.43 |
| | | Slope | 0.016 | -0.193 | 0.226 | 0.897 | - |
| | ASD-Receptive Language | Intercept | -1.234 | -4.259 | 1.790 | 0.502 | - |
| | | Slope | -0.137 | -0.409 | 0.135 | 0.409 | - |
| | TD-Receptive Language | Intercept | 6.82 | 3.96 | 9.67 | < .001* | 0.72 |
| | | Slope | -0.184 | -0.396 | 0.028 | 0.153 | - |
| | ASD-Expressive Language | Intercept | 1.736 | -0.749 | 4.221 | 0.250 | - |
| | | Slope | -0.059 | -0.328 | 0.210 | 0.719 | - |
| | TD-Expressive Language | Intercept | 3.73 | 1.18 | 6.29 | 0.016* | 0.36 |
| | | Slope | -0.037 | -0.248 | 0.175 | 0.777 | - |

*Note. Estimate* = regression coefficient, * = $P < .05$. Please note that only significant estimates (i.e., intercepts and slopes) are presented in this table. All the tests are two-sided. R & R Behavior = Restricted & Repetitive Behavior. ABC = Adaptive Behavior Composite. CI: 95% Confidence Interval.

# Reporting Summary

## Statistics

For all statistical analyses, confirm that the following items are present in the figure legend, table legend, main text, or Methods section.

| n/a | Confirmed | |
|---|---|---|
| ☒ | ☐ | The exact sample size (*n*) for each experimental group/condition, given as a discrete number and unit of measurement |
| ☒ | ☐ | A statement on whether measurements were taken from distinct samples or whether the same sample was measured repeatedly |
| ☐ | ☒ | The statistical test(s) used AND whether they are one- or two-sided *Only common tests should be described solely by name; describe more complex techniques in the Methods section.* |
| ☐ | ☒ | A description of all covariates tested |
| ☐ | ☒ | A description of any assumptions or corrections, such as tests of normality and adjustment for multiple comparisons |
| ☐ | ☒ | A full description of the statistical parameters including central tendency (e.g. means) or other basic estimates (e.g. regression coefficient) AND variation (e.g. standard deviation) or associated estimates of uncertainty (e.g. confidence intervals) |
| ☐ | ☒ | For null hypothesis testing, the test statistic (e.g. *F*, *t*, *r*) with confidence intervals, effect sizes, degrees of freedom and *P* value noted *Give P values as exact values whenever suitable.* |
| ☒ | ☐ | For Bayesian analysis, information on the choice of priors and Markov chain Monte Carlo settings |
| ☐ | ☒ | For hierarchical and complex designs, identification of the appropriate level for tests and full reporting of outcomes |
| ☐ | ☒ | Estimates of effect sizes (e.g. Cohen's *d*, Pearson's *r*), indicating how they were calculated |
| | | *Our web collection on statistics for biologists contains articles on many of the points above.* |

## Software and code

Policy information about availability of computer code

| Data collection | The study data collection was based on 1-Year Well Baby Check Up Approach (Pierce et al., 2011) and Get SET Early model (Pierce et al., 2021). |
|---|---|
| Data analysis | All the data cleaning and the majority of analyses were conducted in R and Longitudinal analyses were conducted using Mplus version 8.3. |

For manuscripts utilizing custom algorithms or software that are central to the research but not yet described in published literature, software must be made available to editors and reviewers. We strongly encourage code deposition in a community repository (e.g. GitHub). See the Nature Portfolio guidelines for submitting code & software for further information.

## Data

Policy information about availability of data

All manuscripts must include a data availability statement. This statement should provide the following information, where applicable:
- Accession codes, unique identifiers, or web links for publicly available datasets
- A description of any restrictions on data availability
- For clinical datasets or third party data, please ensure that the statement adheres to our policy

The data supporting the findings of this study are available on the ACE lab's GitHub page. Link: https://github.com/ACE-UCSD/Autism-Sex-Differences-Analysis-Pathway.

# Research involving human participants, their data, or biological material

Policy information about studies with [human participants or human data](). See also policy information about [sex, gender (identity/presentation), and sexual orientation]() and [race, ethnicity and racism]().

| | |
|---|---|
| Reporting on sex and gender | The findings of this study are applicable solely to sex, as investigating the sex differences in toddlers with autism was the primary objective. Our data stratify all subjects by sex, which is indicated in a separate column on the primary datasheet. It should be noted that, because the study involves toddlers, sex was determined based on parental reports. Consent has been obtained to share individual-level data that has been de-identified. |
| Reporting on race, ethnicity, or other socially relevant groupings | All of the data collected in the current study was funded by national and state health agencies, and the categorizations of race and ethnicity are based on NIH guidelines. |
| Population characteristics | Please see Table 1 in the manuscript for detailed information regarding population characteristics. |
| Recruitment | Approximately 75% of the sample was collected using a general population-based screening approach called Get Set Early which minimizes recruitment biases because all toddlers are screened with parent report questions at pediatric well-baby check ups at 12:18 and 24 months. |
| Ethics oversight | This research has met all the ethical requirements regarding human research protection program under the approval of the UCSD Office of IRB administration (project number: 202115). |

Note that full information on the approval of the study protocol must also be provided in the manuscript.

# Field-specific reporting

Please select the one below that is the best fit for your research. If you are not sure, read the appropriate sections before making your selection.

☐ Life sciences   ☒ Behavioural & social sciences   ☐ Ecological, evolutionary & environmental sciences

For a reference copy of the document with all sections, see [nature.com/documents/nr-reporting-summary-flat.pdf]()

# Behavioural & social sciences study design

All studies must disclose on these points even when the disclosure is negative.

| | |
|---|---|
| Study description | The study is quantitative and includes cross-sectional, cluster, and longitudinal analyses. |
| Research sample | A total of 2,618 toddlers participated and included those with a diagnosis of ASD (N=1539; 1200 M and 339 F; mean age 28.6 months), DD (N=478; 349 M and 129 F; mean age 26.0 months) as well as those who were TD (N=601; 349 M and 252 F; mean age 25.7 months). |
| Sampling strategy | The study data collection was based on 1-Year Well Baby Check Up Approach (Pierce et al., 2011) and Get SET Early model (Pierce et al., 2021). |
| Data collection | The study data collection was based on 1-Year Well Baby Check Up Approach (Pierce et al., 2011) and Get SET Early model (Pierce et al., 2021). |
| Timing | 2002-2022 |
| Data exclusions | Data were excluded based on criteria such as deafness, blindness, or less than 50% exposure to English or Spanish. |
| Non-participation | Data from families who declined to participate or dropped out of the study were not included in the analysis. Since this study encompasses a combination of data collected over an 11-year period, tracking can only be reported approximately. We estimate that less than one percent of the participating families dropped out of the study. |
| Randomization | Participants were assigned to groups based on their sex and diagnosis, in alignment with the study's objectives. |

# Reporting for specific materials, systems and methods

We require information from authors about some types of materials, experimental systems and methods used in many studies. Here, indicate whether each material, system or method listed is relevant to your study. If you are not sure if a list item applies to your research, read the appropriate section before selecting a response.

## Materials & experimental systems

| n/a | Involved in the study |
|-----|----------------------|
| ☒ ☐ | Antibodies |
| ☒ ☐ | Eukaryotic cell lines |
| ☒ ☐ | Palaeontology and archaeology |
| ☒ ☐ | Animals and other organisms |
| ☒ ☐ | Clinical data |
| ☒ ☐ | Dual use research of concern |
| ☒ ☐ | Plants |

## Methods

| n/a | Involved in the study |
|-----|----------------------|
| ☒ ☐ | ChIP-seq |
| ☒ ☐ | Flow cytometry |
| ☒ ☐ | MRI-based neuroimaging |

## Plants

| | |
|---|---|
| Seed stocks | *Report on the source of all seed stocks or other plant material used. If applicable, state the seed stock centre and catalogue number. If plant specimens were collected from the field, describe the collection location, date and sampling procedures.* |
| Novel plant genotypes | *Describe the methods by which all novel plant genotypes were produced. This includes those generated by transgenic approaches, gene editing, chemical/radiation-based mutagenesis and hybridization. For transgenic lines, describe the transformation method, the number of independent lines analyzed and the generation upon which experiments were performed. For gene-edited lines, describe the editor used, the endogenous sequence targeted for editing, the targeting guide RNA sequence (if applicable) and how the editor was applied.* |
| Authentication | *Describe any authentication procedures for each seed stock used or novel genotype generated. Describe any experiments used to assess the effect of a mutation and, where applicable, how potential secondary effects (e.g. second site T-DNA insertions, mosiacism, off-target gene editing) were examined.* |

