## [Peer Review File · Nature Human Behaviour]

Large Scale Examination of Early-Age Sex Differences in Neurotypical, Autism Spectrum Disorder, and Toddlers with Other Developmental Conditions

Corresponding Author: Dr Karen Pierce

Version 0:

Decision Letter:

10th May 2024

Dear Dr Pierce,

Thank you once again for your manuscript, entitled "Large Scale Examination of Early-Age Sex Differences in Neurotypical, Autism Spectrum Disorder, and Toddlers with Other Developmental Conditions," and many thanks for your patience during the peer review process. Please accept our sincere apologies for the delay in reaching a decision.

Your manuscript has now been evaluated by 2 reviewers, whose comments are included at the end of this letter. Although the reviewers find your work to be of interest, they also raise some important concerns. We are very interested in the possibility of publishing your study in Nature Human Behaviour, but would like to consider your response to these concerns in the form of a revised manuscript before we make a decision on publication.

To guide the scope of the revisions, the editors discuss the referee reports in detail within the team, including with the chief editor, with a view to (1) identifying key priorities that should be addressed in revision and (2) overruling referee requests that are deemed beyond the scope of the current study. We hope that you will find the prioritised set of referee points to be useful when revising your study. Please do not hesitate to get in touch if you would like to discuss these issues further.

1. Our reviewers are generally positive about your work but raise a number of questions regarding the analytical choices. Please carefully address these concerns in your revision.
2. Both reviewers find that a clearer motivation for the focus on early childhood differences is warranted. We agree with the reviewers and ask you to address this concern in full.
3. Finally, please discuss limitations of your study and tone down claims regarding the implications of your findings for the extreme male brain theory.

In sum, we invite you to revise your manuscript taking into account all reviewer and editor comments. We are committed to providing a fair and constructive peer-review process. Do not hesitate to contact us if there are specific requests from the reviewers that you believe are technically impossible or unlikely to yield a meaningful outcome.

We hope to receive your revised manuscript within two months. I would be grateful if you could contact us as soon as possible if you foresee difficulties with meeting this target resubmission date.

- Include a "Response to the editors and reviewers" document detailing, point-by-point, how you addressed each editor and referee comment. If no action was taken to address a point, you must provide a compelling argument. When formatting this document, please respond to each reviewer comment individually, including the full text of the reviewer comment verbatim followed by your response to the individual point. This response will be used by the editors to evaluate your revision and sent back to the reviewers along with the revised manuscript.
- Highlight all changes made to your manuscript or provide us with a version that tracks changes.

Link Redacted

We look forward to seeing the revised manuscript and thank you for the opportunity to review your work. Please do not hesitate to contact me if you have any questions or would like to discuss these revisions further.

Sincerely,

██████████
██████████ PhD
██████████
Nature Human Behaviour

Reviewer expertise:

Reviewer #1: ASD ; Similarity Network Fusion ; cluster analysis

Reviewer #2: ASD ; cluster analysis

REVIEWER COMMENTS:

Reviewer #1:
Remarks to the Author:

The manuscript by Nazari et al. examines for sex differences among clinical/behavioural variables sampled in toddlers with and without autism in a large community sample collected using a population screening method implemented across pediatric clinics over a 20 year period in San Diego (n~2600: ~1500 autism, ~600TD, ~470 DD). This study found limited cross-sectional and longitudinal sex differences in various clinical scores between male and female toddlers with autism, and greater sex differences between male and female typically developing toddlers. Data integration and clustering using SNF based on inclusion of three data types (ADOS, vineland, MSEL) in a subset of the sample (~1600 Autism/TD) generally divided the sample into those with low/med/high scores on these domains, regardless of clinical diagnosis, and sex (sex composition was similar bw M and F across data-driven subgroups identified). This study is important as it adds to an existing literature that is limited by sample size and ascertainment approaches, and indicates that there are minimal sex differences across symptom/cognitive/language/motor/eye tracking domains among toddlers screening positive and referred for developmental assessment or those referred due to developmental concerns in the San Diego area. The paper is very well-written and the introduction provides a strong claim as to why this study is important. The methods conducted in the study adequately answer the questions posed and incorporate a large cross-sectional sample with longitudinal data.

Some comments/suggestions are included below:

Main Comments:

1. The study uses a county-wide ascertainment approach whereby children who met a cut off on a screening tool were referred for further assessment. Can the authors mention the sensitivity/specificity data for the CSBS IT? Is it possible that this screening method is less sensitive to detection of autism in younger girls (which is suggested by the sex difference findings in the screening tool itself, though effect sizes are very small). The children being picked up using this method are likely to have more severe autism presentations, given early developmental signs present (ie these data may not extend to children with less severe presentations that are not diagnosed until much later). This could be mentioned as a limitation in the discussion and possibly that while this county wide screening approach is an improvement over other ascertainment approaches, ideally replication in other geographic locations using other measures for screening are included for external validation of current results. The other info to add is whether all the children in this study are being provided with private/insurance covered medical care ?

2. The male-female*group comparisons clearly show that there are minimal overall differences bw ASD toddlers and more cross-domain differences are present among TDs. SNF findings show a range of ability across the ASD sample, with many children being subgrouped with TD children and equal proportions in each cluster for males and females. This is interesting and indicates that the sample has a large range of ability across variables of interest, even among children picked up in very early screening as toddlers. I did wonder why sex was not included as an input feature in SNF? SNF will identify participant similarity based on the information provided (ie input features included). Given there are minimal differences bw sexes for behavioural scales, it is not surprising that SNF results do not cluster based on sex. I do wonder what the result would be if sex is inputted? This may be an interesting addition to the cross-sectional analysis.

3. How correlated are the external validators to those in the model? Can a correlation matrix among the full sample with all variables of interest included be provided? If external validation measures are highly correlated with input features than they provide additional information about clusters, but are not serving as an external validation.
4. I found the cluster visualization a little difficult to follow – is it possible to colour/format data points according to diagnosis and sex, in order to better see how participants are clustered across sex and diagnosis. Is it also possible to provide a pie chart indicating the number of autistic M/F, TD M/F that are included in each cluster to complement information provided in table format?
5. The introduction could benefit from discussing how studying sex differences at such an early timepoint can inform hypotheses re trajectories and development. Currently, the introduction does an excellent job at detailing the inconsistent results on ASD sex differences in clinical symptoms. However, somewhere in intro or discussion it would be helpful to emphasize that these results may not extend to older children (ie not picked up early). It would also be helpful to add some clinical interpretation. That is, current concerns are that screening/ASD measures may be less sensitive to more subtle autism presentations in females. How do these findings allay these concerns in toddlers, and would this extend to older females?
6. As shown in Table 1, the sample does come from a higher-SES group in which the median income in the sample is higher than the median income in the US (hence question re whether receiving are through private insurance covered services). Can the authors reflect more on this in the discussion re whether SES may potentially influence sex-specific/nonspecific findings in some way?
7. The authors currently note that the Extreme Male Brain theory is supported by current findings. It is clear that TD F outperform M across many domains. However, can another visualization be added to show baseline performance side by side (ie across ASD and TD groups by sex) with confidence intervals? Based on the performance data included in tables, it appears that while performance looks overall similar wrt means in TD M and F, range of scores appears greater in males, suggesting more TD males have scores that are on lower end of the distribution. Would be helpful to visualize this for readers, if it is the case.

Reviewer #2:

Remarks to the Author:

I appreciate the opportunity to review this manuscript, which focused on sex differences in a large dataset of children screened in early childhood for ASD and followed longitudinally. Strengths of this study include a large dataset, comparison of behavior across a variety of relevant domains (i.e., not just autism symptoms). While I found the results interesting, I had significant concerns and confusion around the subgrouping analysis and believe the implications are overly stated. Please see my suggestions for revision below.

In the introduction, the authors discuss past research that has shown variable levels of sex differences. Many of these studies have specifically addressed the developmental nature of sex differences – with more differences between autistic males and females in later childhood and adolescence. I think this is worth noting, and also highlights a question as to why the authors chose to focus on the early childhood period?

It appears that most of your group-level comparisons and cluster analyses only included ASD and TD participants, not those in the DD group. What was the rationale for this? I imagine that including a clinically heterogeneous group would bolster the generalizability of your results.

It also seemed weird to me to break apart your examinations of sex differences within and across clusters to the ASD and TD groups separately. Isn't the assumption of SNF that if you're inputting data into a subgrouping model that subgroups will be comparable, and should not be separated into separate diagnostic groups?

Please provide justification for comparing ADOS scores across males and females in the TD group? The ADOS is meant to differentiate ASD from non-spectrum and does not show utility in capturing dimensional variability in SC or RRB in TD children. Specifically, your longitudinal sex difference analysis does not seem very meaningful in the TD group.

Please provide more information on which version and module of the ADOS was given – did that impact results? The raw scores are very different across modules, and it seems unusual to compare raw scores over such a wide age range and language level.

Please provide more information on what timepoint was used for the sex difference analysis (presented in figure 1).

I was confused on how you identified selection features for the SNF, as well as the selection for the “external” variables. Generally, you shouldn't be comparing groups on selection features, other than to describe your subgroups, since you will obviously find group differences. It seems like the external variables were selected out from the original SNF analysis because they were highly correlated with the selection measures. So I wouldn't use those as validation either.

Pg 4, line 139: Please provide justification for your choice of a 3-cluster solution vs. a 2-cluster solution when both performed equally well.

In the discussion, you note that there are “virtually no” differences between males and females with ASD. That is true, with the exception of the screening instruments (CSBS), which you note briefly in the next paragraph. It seems relevant to note that the screening instruments are not picking up on concerns as well in females who went on to receive a diagnosis, and could contribute to the lack of clinical referrals for females. Please elaborate more on the implications of this in the second paragraph of the discussion.

I would reduce your discussion of the EMB theory. To truly show evidence of the EMB, you would want to compare TD boys to ASD girls, and the EMB would also suggest sex differences within ASD (since it's just a shift in distribution).

Page 7 line 270: "These results do not support prominent speculations about ASD sex differences." – I would temper this language. First, you're only studying early childhood, where many studies find sex differences that present later. Second, you're using tools that were designed to identify impairment (ADOS, VABS), not capture dimensional variability in symptoms. There still could be nuanced differences that aren't be picked up on by these measures.

The paragraph that starts with "Across the past 20 years" feels unnecessary and like you are simply listing results from past studies.

The discussion currently focuses on the lack of sex differences within the ASD sample. However, they fail to acknowledge the difference in sex ratio of their sample (22% of the ASD sample was female). Combined with the fact that they required participants to meet on the ADOS, it is possible that there is a group of girls who are showing mild symptoms but may truly have ASD or ASD-related concerns. I think it's worth highlighting this limitation or alternative explanation.

Smaller points:

It appears that the authors required that participants meet cutoff on the ADOS. What is the rationale for this? The ADOS does not have perfect sensitivity, so we wouldn't expect all kids with ASD to meet.

What do you mean by "ADOS-certified"? Were your assessors all research reliable on the ADOS?

Pg. 3, line 113: Could you be more specific with the following sentence: "the 113 Communication and Symbolic Behavior Scales Infant-Toddler Checklist (CSBS)32, revealed poorer screen scores in ASD 114 boys than girls;"? I imagine you mean that boys demonstrated lower scores/more difficulties?

Pg. 6, line 228 – you say that your sample was uniformly ascertained, but your methods describe that some kids came from clinical referrals (not ASD screening). I'd reduce how strongly you're claiming uniform ascertainment.

Version 1:

Decision Letter:

30th September 2024

Dear Dr Pierce,

Thank you once again for your revised manuscript, entitled "Large Scale Examination of Early-Age Sex Differences in Neurotypical, Autism Spectrum Disorder, and Toddlers with Other Developmental Conditions," and for your patience during the review process.

Your manuscript has now been evaluated by the same reviewers who evaluated your original manuscript. All reviewer feedback is included at the end of this letter. While Reviewer 1 is satisfied with the revisions, Reviewer 2 has some important outstanding concerns. We remain very interested in the possibility of publishing your study in *Nature Human Behaviour*, but would like to consider your response to these outstanding concerns in the form of a revised manuscript before we make a decision on publication.

In sum, we invite you to revise your manuscript taking into account all reviewer and editor comments. We are committed to providing a fair and constructive peer-review process. Do not hesitate to contact us if there are specific requests from the reviewers that you believe are technically impossible or unlikely to yield a meaningful outcome.

We hope to receive your revised manuscript within 4-8 weeks. I would be grateful if you could contact us as soon as possible if you foresee difficulties with meeting this target resubmission date.

- Include a "Response to the editors and reviewers" document detailing, point-by-point, how you addressed each editor and referee comment. If no action was taken to address a point, you must provide a compelling argument. This response will be used by the editors and reviewers to evaluate your revision.

- Highlight all changes made to your manuscript or provide us with a version that tracks changes.

Link Redacted

We look forward to seeing the revised manuscript and thank you for the opportunity to review your work. Please do not hesitate to contact me if you have any questions or would like to discuss these revisions further.

Sincerely,

██████████ ██████████
██████████ ██████████ PhD
██████████
Nature Human Behaviour

REVIEWER COMMENTS:

Reviewer #1 (Remarks to the Author):

The authors have undertaken a very thorough response to prior reviews. I am satisfied with the response and modifications made and feel the current manuscript makes an important contribution to the literature

Reviewer #2 (Remarks to the Author):

I appreciate the authors' careful consideration of the suggestions from reviewers, and believe the revised manuscript to be much improved. I have a few remaining concerns after reviewing the rebuttal and updated manuscript, as described below.

Additional justification for choosing 3 cluster rather than 2 is still needed. There should be some theoretical justification if there is no statistical reason to choose 3 over 2. Just wanting to look at "more nuanced" differences isn't sufficient of a reason. There will always be more nuanced differences if you are breaking a group up into 3 vs. 2 groups.

I also noticed that it seems like you did not include the ADOS RRB score into your cluster analysis. If that is the case, I imagine that would be a nice "external variable" to compare the groups on.

Additional justification is needed in choices surrounding the ADOS. From the authors' response, it seems like they compared algorithm raw score totals across modules and versions. This is simply not acceptable – the items included and scales are different. If you plan to compare across module, you need to use the CSS. If you want to look at raw scores, you should only look at differences within each version, module, and algorithm set (e.g., no words vs. some words).

I also would strongly advocate against using ADOS scores in the TD group. If you use CSSs as suggested, those were only normed on diagnosed individuals and are not meant to be used in TD groups. Furthermore, there is recent evidence that the ADOS SA factor does not fit well in TD individuals, particularly females (Burrows et al., 2022), highlighting that it is not capturing meaningful variability in TD kids. That paper suggested not using ADOS scores as a dimensional measure for TD kids.

Did you have the same proportion of ASD males and females in the train and test set?

Please include the means and SD for each combination of M/F and ASD/TD for your primary scales of interest. Right now I'm only seeing the test statistic of the differences.

Smaller points:

A smaller ADOS point, but in the manuscript, you describe that you used the ADOS-2, but in the response it says that you had a combination of ADOS-G and ADOS-2. Please check the manuscript for accuracy.

My understanding is that you should never report a p-value of 0.000. It should always say $p < 0.0001$. I defer to the journal editor on this point, but that's the convention I'm aware of.

Please reword the following caption for figure 1. It is currently unclear to me whether males or females had higher scores. It's also hard when the direction of effect (e.g., higher scores equals more impairment vs. higher scores equals better skills in that area) is different across measures. Is there a way to put them on the same scale? "asterisks (*) indicate statistically significant sex differences, with girls outperforming boys in those marked areas."

Version 2:

Decision Letter:

Our ref: NATHUMBEHAV-24010073B

5th December 2024

Dear Dr. Pierce,

Thank you for submitting your revised manuscript "Large Scale Examination of Early-Age Sex Differences in Neurotypical, Autism Spectrum Disorder, and Toddlers with Other Developmental Conditions" (NATHUMBEHAV-24010073B). It has now been seen by the original referees and their comments are below. As you can see, the reviewers find that the paper has improved in revision. We will therefore be happy in principle to publish it in Nature Human Behaviour, pending minor revisions to satisfy the referees' final requests and to comply with our editorial and formatting guidelines.

We are now performing detailed checks on your paper and will send you a checklist detailing our editorial and formatting requirements within two weeks. Please do not upload the final materials and make any revisions until you receive this additional information from us.

Sincerely,

[REDACTED]

[REDACTED] PhD

Nature Human Behaviour

Reviewer #2 (Remarks to the Author):

I appreciate the authors' thorough responses to revision requests, and believe it has resulted in a much improved manuscript. I only have a few minor suggestions for revision remaining.

I appreciate the authors' thorough responses to my comment about ADOS CSS versus raw scores. And I do appreciate the increased variability afforded by using raw scores. If the authors are going to use these scores, I'd appreciate more information about what module was given, and looking at sex differences within modules. Could you add in N's for each module/algorithm in either Table 1 or the supplement? I'd also appreciate a set of figures similar to Figure 3 that has the ADOS SA and RRB scores split by module.

Could you provide a more descriptive title for Extended Table 1? I believe that table is looking at sex differences in different measures, but it's not clear from just looking at the table.

I'd recommend updating the abstract to more comprehensively describe the results of this study.

Version 3:

Decision Letter:

Dear Dr Pierce,

We are pleased to inform you that your Article "Large Scale Examination of Early-Age Sex Differences in Neurotypical, Autism Spectrum Disorder, and Toddlers with Other Developmental Conditions", has now been accepted for publication in Nature Human Behaviour.

Authors may need to take specific actions to achieve [a href="https://www.springernature.com/gp/open-research/funding/policy-compliance-faqs"](https://www.springernature.com/gp/open-research/funding/policy-compliance-faqs) compliance with funder and institutional open access mandates. If your research is supported by a funder that requires immediate open access (e.g. according to [a href="https://www.springernature.com/gp/open-research/plan-s-compliance"](https://www.springernature.com/gp/open-research/plan-s-compliance) Plan S principles) then you should select the gold OA route, and we will direct you to the compliant route where possible. For authors selecting the subscription publication route, the journal's standard licensing terms will need to be accepted, including [a href="https://www.springernature.com/gp/open-research/policies/journal-policies"](https://www.springernature.com/gp/open-research/policies/journal-policies) self-archiving policies. Those licensing terms will supersede any other terms that the author or any third party may assert apply to any version of the manuscript.

Once your manuscript is typeset and you have completed the appropriate grant of rights, you will receive a link to your electronic proof via email with a request to make any corrections within 48 hours. If, when you receive your proof, you cannot meet this

deadline, please inform us at rjsproduction@springernature.com immediately. Once your paper has been scheduled for online publication, the Nature press office will be in touch to confirm the details.

With best regards,

[Redacted]

[Redacted] PhD

[Redacted]
Nature Human Behaviour

P.S. Click on the following link if you would like to recommend Nature Human Behaviour to your librarian
<http://www.nature.com/subscriptions/recommend.html#forms>

** Visit the Springer Nature Editorial and Publishing website at http://editorial-jobs.springernature.com?utm_source=ejp_NHumB_email&utm_medium=ejp_NHumB_email&utm_campaign=ejp_NHumB for more information about our career opportunities. If you have any questions please click [here](mailto:editorial.publishing.jobs@springernature.com).**

Open Access This Peer Review File is licensed under a Creative Commons Attribution 4.0 International License, which permits use, sharing, adaptation, distribution and reproduction in any medium or format, as long as you give appropriate credit to the original author(s) and the source, provide a link to the Creative Commons license, and indicate if changes were made. In cases where reviewers are anonymous, credit should be given to 'Anonymous Referee' and the source. The images or other third party material in this Peer Review File are included in the article's Creative Commons license, unless indicated otherwise in a credit line to the material. If material is not included in the article's Creative Commons license and your intended use is not permitted by statutory regulation or exceeds the permitted use, you will need to obtain permission directly from the copyright holder.

REVIEWER COMMENTS:

Reviewer #1:

Remarks to the Author:

The manuscript by Nazari et al. examines for sex differences among clinical/behavioural variables sampled in toddlers with and without autism in a large community sample collected using a population screening method implemented across pediatric clinics over a 20 year period in San Diego (n~2600: ~1500 autism, ~600TD, ~470 DD). This study found limited cross-sectional and longitudinal sex differences in various clinical scores between male and female toddlers with autism, and greater sex differences between male and female typically developing toddlers. Data integration and clustering using SNF based on inclusion of three data types (ADOS, vineland, MSEL) in a subset of the sample (~1600 Autism/TD) generally divided the sample into those with low/med/high scores on these domains, regardless of clinical diagnosis, and sex (sex composition was similar bw M and F across data-driven subgroups identified). This study is important as it adds to an existing literature that is limited by sample size and ascertainment approaches, and indicates that there are minimal sex differences across symptom/cognitive/language/motor/eye tracking domains among toddlers screening positive and referred for developmental assessment or those referred due to developmental concerns in the San Diego area. The paper is very well-written and the introduction provides a strong claim as to why this study is important. The methods conducted in the study adequately answer the questions posed and incorporate a large cross-sectional sample with longitudinal data.

Some comments/suggestions are included below:

Main Comments:

1. The study uses a county-wide ascertainment approach whereby children who met a cut off on a screening tool were referred for further assessment. Can the authors mention the sensitivity/specificity data for the CSBS IT? Is it possible that this screening method is less sensitive to detection of autism in younger girls (which is suggested by the sex difference findings in the screening tool itself, though effect sizes are very small). The children being picked up using this method are likely to have more severe autism presentations, given early developmental signs present (ie these data may not extend to children with less severe presentations that are not diagnosed until much later). This could be mentioned as a limitation in the discussion and possibly that while this county wide screening approach is an improvement over other ascertainment approaches, ideally replication in other geographic locations using other measures for screening are included for external validation of current results. The other info to add is whether all the children in this study are being provided with private/insurance covered medical care?

Thank you for highlighting this important point. The CSBS IT-Checklist was developed and validated in 2002 by Wetherby and Prizant as a broadband screening tool to detect toddlers with social communication delays in general, not autism per se⁷. The manual, which was based on a validation sample of 2,188 toddlers, notes sensitivity and specificity of 76% and 82% respectively for detecting toddlers with communication delays. Only a few studies exist that examine validation metrics for autism specifically. In 2004, Wetherby and colleagues assessed the sensitivity and specificity of the CSBS IT Checklist using a small sample of toddlers: 18 with ASD, 18 with developmental delays (DD), and 18 typically developing (TD) (from an initial screened sample of 3,026 children). They reported an estimated sensitivity and specificity of 88.9% when the ASD and DD groups were combined, although noted an increase in sensitivity to 94.4% when only the ASD and TD group were compared⁸. The sample size, however, was extremely small and there was a lack of clarity regarding how subjects were selected for the evaluation as well as other issues, suggesting that these results can be taken as preliminary. A few years later the test developers performed a larger study specifically designed to validate the CSBS IT-Checklist as an ASD screening tool, but reported only PPV (70.5%-79%) and NPV (87.9-99%)⁹, not sensitivity or specificity.

The reality is that validation metrics associated with screening tools are challenging to accurately determine in pediatric populations. Sensitivity, for example, is calculated based on a ratio of true positives/true positives+false negatives. In order to accurately determine if a toddler is a 'false negative' would require performing follow up diagnostic evaluations on the entire screened cohort, which often ranges from 10,000-50,000 toddlers, which is not feasible. Medical professionals disproportionately refer toddlers who fail the screen (i.e., they do not refer many who pass) for an evaluation, allowing for potentially more accurate estimates of positive predictive value. It is for this reason that many researchers that use the

CSBS, including our team, provide estimates of PPV (usually around 75%) but do not report other metrics¹⁰. A recent paper, however, addressed the challenges inherent in determining the accuracy of infant screeners head-on within the context of a different screening tool - the M-CHAT¹¹. Specifically, study researchers examined digital medical records of the entire screened cohort to determine diagnostic outcome, and found that sensitivity in particular was reduced relative to what the test developers had originally estimated¹¹. To the best of our knowledge no such comprehensive effort has occurred as it relates to the CSBS.

Overall, we agree with your point that it is unlikely that the sensitivity of the CSBS is 100%, and some ASD toddlers may have been missed by the screening tool. This, as you note, could result in a cohort that could be slightly more symptomatic relative to the ASD population as a whole. However, data from our paper suggests that our cohort is representative of the ASD population and does not have a disproportionate number of highly symptomatic toddlers. For example, Extended Tables 3 & 4 in the Supplement report the standardized expressive and receptive language scores for our ASD cohort based on cluster membership. As illustrated, only a small percentage of toddlers (~2%-18%, depending on the dataset) fall into the severe cluster.

Regarding the sensitivity of the CSBS as it relates to the detection of females with ASD, we do not believe this is a major issue given the fact that the male-female ratio in the current study is actually more strongly in favor of ASD female detection than the national average (3.6 to 1 in the current study, versus 3.8 to 1 in the most recent national average)¹². We now address this key point in the Discussion.

Regarding private insurance/MediCare, since this study is fully grant funded by NIMH, we did not request payment or any insurance information and thus have no knowledge or ability to determine Medicare status.

Please see page 7 in the manuscript:

It is important to consider the possible influence our early ASD subject recruitment approach, *Get SET Early*, which relies on a toddler's failure of the CSBS at well-baby check-ups to prompt a referral to our Center, may have had on study results. Although sex specific norms are not provided by the test developers, it is possible that this screening method is less sensitive to detection of autism in younger girls given that male toddlers with ASD had slightly lower (worse) CSBS scores than females with ASD. However, counter to this is the fact that the male-female ratio in the current study is actually more strongly in favor of ASD female detection than the national average (3.6 to 1 in the current study, versus 3.8 to 1 in the most recent CDC average¹²), suggesting that the *Get SET Early* approach detects females with ASD at expected, or better than expected, rates. Other possible issues relating to the use of the *Get SET Early* model is the fact that it is unlikely that extremely high functioning individuals with ASD will be detected using developmental screening tools, such as the CSBS, that rely on the presence of observable symptoms. Indeed, although precise estimates regarding ASD late identified cases are not formally established, some studies suggest that as many as 6-25% of ASD individuals do not receive a diagnosis until school age or later^{13,14}. It is thus possible that results from the present study may, or may not, generalize to the subset of ASD individuals with mild presentations who do not receive diagnoses until later in life¹⁴⁻¹⁶. Future studies can examine this possibility by examining sex differences in late detected groups in comparison to individuals from more traditionally detected cohorts.

2. The male-female*group comparisons clearly show that there are minimal overall differences bw ASD toddlers and more cross-domain differences are present among TDs. SNF findings show a range of ability across the ASD sample, with many children being subgrouped with TD children and equal proportions in each cluster for males and females. This is interesting and indicates that the sample has a large range of ability across variables of interest, even among children picked up in very early screening as toddlers. I did wonder why sex was not included as an input feature in SNF? SNF will identify participant similarity based on the information provided (ie input features included). Given there are minimal differences bw sexes for behavioural scales, it is not surprising that SNF results do not cluster based on sex. I do wonder what the result would be if sex is inputted? This may be an interesting addition to the cross-sectional analysis.

Thank you for your insightful observations. Regarding your question about including sex as an input feature in SNF, we initially chose not to include it due to the minimal differences observed between sexes on behavioral scales. Additionally, adding sex as an input feature in SNF may result in clusters that are predominantly male or female. Therefore, our aim was to examine sex differences after clustering was done. Nonetheless, to address your suggestion, we conducted an additional SNF analysis with sex included as an input feature. The results showed that incorporating sex had a minimal

effect on the clustering outcome. The clusters remained largely consistent with our previous analysis, indicating that sex did not significantly alter the subgrouping pattern. This outcome supports the observation that sex differences are minimal within the scope of the behavioral scales used. We now include this in the Methods, and results can be found in Supplementary Material.

Please see page 10 in the manuscript:

In addition to the primary SNF analysis, we conducted a separate SNF analysis incorporating sex as a main feature. This allowed us to further examine the effect of sex on the clustering outcomes.

Please see page 33 in the manuscript:

Supplementary Table 15 – SNF with sex as a main feature.

ASD				Social domain		Motor domain		Language domain				
Cluster	Diagnosis	Sex	Age	SA	SOC	MTR	FM	COM	RL	EL	N	
High (C2)	ASD	F	21.8	9.6	95.4	99.2	106.3	95	86.4	80.8	27	13.1%
	ASD	M	20.2	8.1	99	101.3	105.7	94.3	94.1	79.8	73	10.6%
Medium (C3)	ASD	F	25.4	14.9	84.9	91.7	82.3	76.3	48.9	53.6	143	69.4%
	ASD	M	25.9	13.7	82.9	92.1	81.2	75.4	54.8	54.7	512	74.6%
Low (C1)	ASD	F	27.5	17.2	68.3	78.4	61.3	56.4	24.3	35.2	36	17.5%
	ASD	M	29.6	17.8	67.2	80.5	57.3	52.7	21.6	29.9	101	14.7%

TD				Social domain		Moto domain		Language domain				
Cluster	Diagnosis	Sex	Age	SA	SOC	MTR	FM	COM	RL	EL	N	
High (C2)	TD	F	25.4	2.1	105.9	101.7	107.6	104.4	110.9	104.1	183	98.4%
	TD	M	27	2.3	103.4	100	102.1	102.7	105.2	102.1	250	96.5%
Medium (C3)	TD	F	23.1	6	98.3	88.3	87.3	84.7	83.2	67	3	1.6%
	TD	M	27.7	6.9	89.1	91.6	92.4	90.1	85.6	89.9	9	3.5%

Note. SA: ADOS Social Affect, SOC: Vineland Socialization, MTR: Vineland Motor Skills, FM: Std. MSEL Fine Motor, COM: Vineland Communication, RL: Std. MSEL Receptive Language, EL: Std. MSEL Expressive Language. Yellow highlighted cells show sex differences with darker color indicating higher means.

3. How correlated are the external validators to those in the model? Can a correlation matrix among the full sample with all variables of interest included be provided? If external validation measures are highly correlated with input features than they provide additional information about clusters, but are not serving as an external validation.

We understand the importance of ensuring that external validation measures provide independent validation of the clusters. To address your concern, we have included a correlation matrix that shows the relationships between the SNF input features and the external validation variables. This matrix includes all variables of interest for the full SNF sample.

	SNF							External Variables					
	SA	SOC	MTR	FM	COM	RL	EL	RR	TOT	DLY	ABC	VR	%Fixation
SA	1	-0.7	-0.4	-0.6	-0.7	-0.8	-0.8	0.7	1	-0.6	-0.7	-0.7	-0.5
SOC	-0.7	1	0.6	0.6	0.8	0.7	0.7	-0.6	-0.7	0.8	0.9	0.6	0.4
MTR	-0.4	0.6	1	0.5	0.6	0.4	0.4	-0.4	-0.4	0.7	0.7	0.5	0.3
FM	-0.6	0.6	0.5	1	0.6	0.7	0.6	-0.6	-0.6	0.6	0.7	0.7	0.4
COM	-0.7	0.8	0.6	0.6	1	0.8	0.8	-0.7	-0.7	0.8	0.9	0.7	0.5
RL	-0.8	0.7	0.4	0.7	0.8	1	0.8	-0.7	-0.8	0.6	0.8	0.7	0.5
EL	-0.8	0.7	0.4	0.6	0.8	0.8	1	-0.7	-0.8	0.6	0.8	0.7	0.4
RR	0.7	-0.6	-0.4	-0.6	-0.7	-0.7	-0.7	1	0.9	-0.6	-0.7	-0.6	-0.4
TOT	1	-0.7	-0.4	-0.6	-0.7	-0.8	-0.8	0.9	1	-0.6	-0.7	-0.7	-0.5
DLY	-0.6	0.8	0.7	0.6	0.8	0.6	0.6	-0.6	-0.6	1	0.9	0.6	0.4
ABC	-0.7	0.9	0.7	0.7	0.9	0.8	0.8	-0.7	-0.7	0.9	1	0.7	0.4
VR	-0.7	0.6	0.5	0.7	0.7	0.7	0.7	-0.6	-0.7	0.6	0.7	1	0.4
%Fixation	-0.5	0.4	0.3	0.4	0.5	0.5	0.4	-0.4	-0.5	0.4	0.4	0.4	1

In this matrix, yellow cells show external variable correlations to the main features in SNF. As the reviewer mentioned, some of the external validation measures were highly correlated with the SNF input features. To ensure the robustness of our validation, we selected new variables that were not included in our original analysis (see below). Specifically, we used subscales within the CDI-WG (Words and Gestures) and CDI-WS (Words and Sentences) as new external validation variables. These variables demonstrated lower correlations with the primary features used in the SNF analysis, thereby providing more independent validation of our clustering results. Moreover, cluster separation analyses continue to reveal significant differences between clusters with large effect sizes. We now include this in our updated Supplementary Table 8 and 9.

	SA	SOC	MTR	FM	COM	RL	EL	%Fixation	WG-Words Produced	WG-Words Understood	WG-Early Gestures	WG-Later Gestures	WG-Total Gestures	WS-Words Produced
SA	1	-0.7	-0.4	-0.6	-0.7	-0.8	-0.8	-0.5	-0.2	-0.3	-0.4	-0.3	-0.4	-0.5
SOC	-0.7	1	0.6	0.6	0.8	0.7	0.7	0.4	0.2	0.3	0.4	0.4	0.4	0.5
MTR	-0.4	0.6	1	0.5	0.6	0.4	0.4	0.3	0.1	0.2	0.3	0.3	0.4	0.4
FM	-0.6	0.6	0.5	1	0.6	0.7	0.6	0.4	0.1	0.1	0.3	0.2	0.2	0.4
COM	-0.7	0.8	0.6	0.6	1	0.8	0.8	0.5	0.3	0.4	0.5	0.4	0.4	0.7
RL	-0.8	0.7	0.4	0.7	0.8	1	0.8	0.5	0.3	0.3	0.4	0.4	0.4	0.6
EL	-0.8	0.7	0.4	0.6	0.8	0.8	1	0.4	0.3	0.3	0.4	0.3	0.4	0.7

Please see page 29 in the manuscript:

Supplementary Table 8. Cluster separation across external variables.

External Variable	Effect	F-ratio	DFn	DFd	GES	P-value
CDI-WG-Words Produced	Clusters	9.05	2	369	0.05	0.000*
CDI-WG-Words Understood	Clusters	12.28	2	369	0.06	0.000*
CDI-WG-Early Gestures	Clusters	35.15	2	369	0.16	0.000*
CDI-WG-Later Gestures	Clusters	17.85	2	369	0.09	0.000*
CDI-WG-Total Gestures	Clusters	26.38	2	369	0.13	0.000*
CDI-WS-Words Produced	Clusters	154.32	2	609	0.34	0.000*
Geopref-Social % Fixation	Clusters	115.92	2	950	0.20	0.000*

Note. * = $P < .05$, *DFn* = Degrees of freedom for numerator, *DFd* = Degrees of freedom for denominator, *GES* = Generalized Eta Squared (effect size).

Supplementary Table 9. Cluster pairwise comparisons in external variables.

Eternal Variable	Cluster		
		1	2
		P-value	P-value
CDI-WG-Words Produced	2	0.000*	NA
	3	0.017*	0.379
CDI-WG-Words Understood	2	0.000*	NA
	3	0.037*	0.407
CDI-WG-Early Gestures	2	0.000*	NA
	3	0.003*	0.301
CDI-WG-Later Gestures	2	0.000*	NA
	3	0.004*	0.047*
CDI-WG-Total Gestures	2	0.000*	NA
	3	0.001*	0.132
CDI-WS-Words Produced	2	0.000*	NA
	3	0.000*	0.000*
Geopref-Social % Fixation	2	0.000*	NA
	3	0.000*	0.000*

Note. Multiple pairwise comparisons were corrected by FDR, * = $P < .05$.

4. I found the cluster visualization a little difficult to follow – is it possible to colour/format data points according to diagnosis and sex, in order to better see how participants are clustered across sex and diagnosis. Is it also possible to provide a pie chart indicating the number of autistic M/F, TD M/F that are included in each cluster to complement information provided in table format?

Sure. Please see below. We now replace the original figure with below.

Please see page 17 in the manuscript:

5. The introduction could benefit from discussing how studying sex differences at such an early timepoint can inform hypotheses re trajectories and development. Currently, the introduction does an excellent job at detailing the inconsistent results on ASD sex differences in clinical symptoms. However, somewhere in intro or discussion it would be helpful to emphasize that these results may not extend to older children (ie not picked up early). It would also be helpful to add some clinical interpretation. That is, current concerns are that screening/ASD measures may be less sensitive to more subtle autism presentations in females. How do these findings allay these concerns in toddlers, and would this extend to older females?

We agree that this is an important consideration. In the original manuscript we wrote:

Clarity on this topic can lead to improved early-age detection and diagnostic procedures; reveal new insights into causes/mechanisms; point to more efficacious early-age treatment protocols; aid in differential planning and targets for intervention; and more generally provide enhanced sex/gender equity in society¹⁷.

To this we now add:

Please see page 3 in the manuscript:

Examination of early age sex differences within the context of longitudinal data in particular can generate important insights into sex-specific developmental trajectories, benefiting parents and clinicians in understanding progression and course.

We also substantially revised the Discussion to address some of your other points including comments regarding sensitivity of screening, and whether or not results extend to older females. Please see revised Discussion on page 6 and 7.

6. As shown in Table 1, the sample does come from a higher-SES group in which the median income in the sample is higher than the median income in the US (hence question re whether receiving care through private insurance covered services). Can the authors reflect more on this in the discussion re whether SES may potentially influence sex-specific/nonspecific findings in some way?

Although our sample does have a median income that is higher than the US population overall, it includes families in both the low and high SES extremes, and is overall in alignment with the San Diego region. In general, enhanced wealth often leads to better healthcare and nutrition, enrichment activities, and increased parent involvement, factors that can all impact brain development¹⁸. The degree to which wealth could mitigate sex differences in ASD specifically, however, is not clear. To begin to consider this question, we stratified our sample into high and low SES (based on a median split) and examined potential sex differences within the low and high SES groups. Our low and high stratified SES analyses did not reveal ASD sex differences in symptom severity based on the ADOS, but did find small differences in two subscales on the Vineland and one subscale on the Mullen, but in opposite directions (i.e., sex differences were found only in the low-income group on the Vineland, while sex differences were found only in the high-income group on the Mullen). Since the study was not designed a priori to examine SES, and results were not compelling (small effect sizes, all Epsilon squared ~ 0.005) or clear, we felt that adding in speculation regarding SES might add an unnecessary layer of complication and confusion to our otherwise clear manuscript. As such we elected not to Discuss it, but can add this information if you would like. Just let us know.

7. The authors currently note that the Extreme Male Brain theory is supported by current findings. It is clear that TD F outperform M across many domains. However, can another visualization be added to show baseline performance side by side (ie across ASD and TD groups by sex) with confidence intervals? Based on the performance data included in tables, it appears that while performance looks overall similar wrt means in TD M and F, range of scores appears greater in males, suggesting more TD males have scores that are on lower end of the distribution. Would be helpful to visualize this for readers, if it is the case.

Please see below the baseline performance with confidence intervals across groups and sex.

Reviewer #2:

Remarks to the Author:

I appreciate the opportunity to review this manuscript, which focused on sex differences in a large dataset of children screened in early childhood for ASD and followed longitudinally. Strengths of this study include a large dataset, comparison of behavior across a variety of relevant domains (i.e., not just autism symptoms). While I found the results interesting, I had significant concerns and confusion around the subgrouping analysis and believe the implications are overly stated. Please see my suggestions for revision below.

In the introduction, the authors discuss past research that has shown variable levels of sex differences. Many of these studies have specifically addressed the developmental nature of sex differences – with more differences between autistic males and females in later childhood and adolescence. I think this is worth noting, and also highlights a question as to why the authors chose to focus on the early childhood period?

This is an important point. We highlighted later studies with sex differences and talked about the focus on the early childhood period in the introduction.

Please see page 3 in the manuscript:

There is now considerable evidence suggesting that ASD begins during prenatal life¹⁻³. Within this context, examination of sex differences at the earliest ages possible is essential given the large impact of very early experience on phenotypic expression⁴. For example, in one study changes in the quality of the home environment dramatically influenced cognitive profiles in infants originally placed in an orphanage by as much as 15 IQ points⁵. In another study, the receptive language ability of children with autism was significantly improved if exposure to a particular intervention occurred at or prior to 18 months in age⁶. While studying extremely young samples does not entirely eliminate the confound of experience, it

provides a snapshot of ASD more proximal to the disorder’s onset, which may generate results that are more biologically, and less experientially, driven. In contrast, studies of older children and adults, while important for understanding the condition and complimentary to studies with infants and toddlers, can not necessarily disentangle the influence of experience. However, more sex differences are reported in studies involving older children¹⁹⁻²², highlighting the importance of investigating sex differences at early ages.

It appears that most of your group-level comparisons and cluster analyses only included ASD and TD participants, not those in the DD group. What was the rationale for this? I imagine that including a clinically heterogeneous group would bolster the generalizability of your results.

Thank you for your point. The rationale for initially focusing primarily on ASD and TD is both statistical, and theoretical. Statistically, to understand ASD in contrast to neurotypical development, we aimed to have a precisely age-matched TD group using a method known as cardinality matching. This method ensures that the groups are comparable on covariates while maximizing the number of matched pairs. Including DD toddlers at this initial stage would have significantly reduced our final sample size because DD is our smallest group (N=478). Theoretically, our DD group is quite heterogeneous, encompassing toddlers with language delay, global developmental delay, and other delays. We didn’t want to overly emphasize this contrast initially but aimed to be thorough by considering this group in a follow-up analysis. In future examinations of this issue, we would curate a smaller, more homogenous sample.

It also seemed weird to me to break apart your examinations of sex differences within and across clusters to the ASD and TD groups separately. Isn’t the assumption of SNF that if you’re inputting data into a subgrouping model that subgroups will be comparable, and should not be separated into separate diagnostic groups?

To address your concern, we conducted an additional SNF analysis exclusively with the ASD group. The results from this analysis were consistent with those obtained from the combined ASD and TD groups. In this SNF, with no TD individuals, we observed a higher percentage of ASD children in the high cluster and a lower percentage in the medium cluster. The percentage of ASD children in the low cluster remained almost similar.

ASD	Sex	Age	Social domain		Motor domain		Language domain			N	
			SA	SOC	MTR	FM	COM	RL	EL		
All Data	F	25.3	14.6	83.4	90.4	81.8	75.3	49.5	53.9	206	
	M	25.8	13.7	82.3	91.4	80.3	74.0	54.1	53.8	686	%
High (C2)	F	22.8	12.1	94.1	98.8	94.3	88.0	67.7	66.4	74	35.9%
	M	23.0	11.1	93.4	99.7	94.1	88.1	76.3	69.2	224	32.7%
Medium (C1)	F	26.6	15.7	79.7	87.6	78.0	71.2	42.9	50.0	108	52.4%
	M	26.7	14.5	79.1	89.0	77.3	70.4	47.8	49.7	390	56.9%
Low (C3)	F	27.2	17.7	67.1	76.6	60.1	54.6	23.2	33.5	24	11.7%
	M	30.2	17.9	65.3	78.5	53.8	49.8	18.8	27.4	72	10.5%

Please provide justification for comparing ADOS scores across males and females in the TD group? The ADOS is meant to differentiate ASD from non-spectrum and does not show utility in capturing dimensional variability in SC or RRB in TD children. Specifically, your longitudinal sex difference analysis does not seem very meaningful in the TD group.

We appreciate your point regarding the primary purpose of the ADOS, which is to differentiate ASD from non-spectrum individuals. However, we included the ADOS scores for the TD group to explore any potential subtle differences in social affect (SA) and restricted and repetitive behaviors (RRB) that might be present, even within the typical development (TD) population. This was done to provide a more comprehensive understanding of these behaviors across both diagnostic groups. Additionally, our primary focus was on the ASD group, but the inclusion of the TD group allowed us to contrast these scores and highlight any differences or similarities. This context is valuable for a comparative understanding of how these behaviors might manifest differently or similarly in males and females, even in a non-clinical population.

Please provide more information on which version and module of the ADOS was given – did that impact results? The raw scores are very different across modules, and it seems unusual to compare raw scores over such a wide age range and language level.

We used ADOS Generic and ADOS-2, including the Toddler Module, Module 1, and Module 2. Only 20.7% of the sample was based on ADOS Generic. In addition, we used algorithm totals for SA/COSO, RR, and Overall Total scores. The total scores could go up to 26 in ADOS Generic and up to 28 in ADOS-2, but this 2-point difference did not affect the results related to sex differences.

Please provide more information on what timepoint was used for the sex difference analysis (presented in figure 1).

Please see page 9 in the manuscript:

Examination of all available data revealed a statistically significant ($p = .000$), 4-month age difference between ASD (mean age 29.4 months) and TD toddlers (mean age 25.2 months), except for the CDI WS test. In order to ensure that any reported differences were not driven by age effects, primary analyses included data from all the time-points available, which ranged from 1 to 5 and then Cardinality Matching (CM)²³ was conducted to achieve an evenly age-matched sample.

I was confused on how you identified selection features for the SNF, as well as the selection for the “external” variables. Generally, you shouldn’t be comparing groups on selection features, other than to describe your subgroups, since you will obviously find group differences. It seems like the external variables were selected out from the original SNF analysis because they were highly correlated with the selection measures. So I wouldn’t use those as validation either.

Please see pages 9 and 10 in the manuscript:

We employed Pearson correlation, a filter method in feature selection techniques, to identify features for inclusion in the SNF analysis. We calculated all possible pairwise correlations among the subscales of ADOS, Vineland, and MSEL, and then removed those subscales that were highly correlated with others. Additionally, we grouped similar subscales that approximately measure the same concept into a separate layer. Therefore, data from three clinical domains including social, language and motor, were considered as three layers in SNF. Within the social domain, we considered ADOS social affect as well as Vineland socialization subscales. For the language domain, we included MSEL receptive language, MSEL expressive language, and Vineland communication subscales. Lastly, the motor domain included Vineland motor skills and MSEL fine motor.

Regarding the selection of external variables, we agree that they should not be highly correlated with the primary features used in the SNF analysis to avoid redundancy. To address your concern, we excluded these variables and instead used other variables such as CDI-WG (Words and Gestures) and CDI-WS (Words and Sentences) subscales, which have lower correlations with the primary features in the SNF analysis.

Updated tables:

Please see page 29 in the manuscript:

Supplementary Table 8. Cluster separation across external variables.

External Variable	Effect	F-ratio	DFn	DFd	GES	P-value
CDI-WG-Words Produced	Clusters	9.05	2	369	0.05	0.000*
CDI-WG-Words Understood	Clusters	12.28	2	369	0.06	0.000*
CDI-WG-Early Gestures	Clusters	35.15	2	369	0.16	0.000*
CDI-WG-Later Gestures	Clusters	17.85	2	369	0.09	0.000*
CDI-WG-Total Gestures	Clusters	26.38	2	369	0.13	0.000*
CDI-WS-Words Produced	Clusters	154.32	2	609	0.34	0.000*
Geopref-Social % Fixation	Clusters	115.92	2	950	0.20	0.000*

Note. * = $P < .05$, *DFn* = Degrees of freedom for numerator, *DFd* = Degrees of freedom for denominator, *GES* = Generalized Eta Squared (effect size).

Supplementary Table 9. Cluster pairwise comparisons in external variables.

Eternal Variable	Cluster	1	2
		P-value	P-value
CDI-WG-Words Produced	2	0.000*	NA
	3	0.017*	0.379
CDI-WG-Words Understood	2	0.000*	NA
	3	0.037*	0.407
CDI-WG-Early Gestures	2	0.000*	NA
	3	0.003*	0.301
CDI-WG-Later Gestures	2	0.000*	NA
	3	0.004*	0.047*
CDI-WG-Total Gestures	2	0.000*	NA
	3	0.001*	0.132
CDI-WS-Words Produced	2	0.000*	NA
	3	0.000*	0.000*
Geopref-Social % Fixation	2	0.000*	NA
	3	0.000*	0.000*

Note. Multiple pairwise comparisons were corrected by FDR, * = $P < .05$.

Pg 4, line 139: Please provide justification for your choice of a 3-cluster solution vs. a 2-cluster solution when both performed equally well.

Please see page 4 in the manuscript:

Based on our experience with clustering ASD and TD children using SNF, the first obvious result of SNF would be a 2-cluster solution, as ASD and TD children tend to separate easily based on their scores. However, in this study, we opted for a 3-cluster solution, which resulted in high, medium, and low clusters, two of them containing both ASD and TD individuals. This approach allowed us to capture a broader range of heterogeneity within the sample, providing more detailed and nuanced insights into the varying characteristics and different levels of traits.

In the discussion, you note that there are “virtually no” differences between males and females with ASD. That is true, with the exception of the screening instruments (CSBS), which you note briefly in the next paragraph. It seems relevant to note that the screening instruments are not picking up on concerns as well in females who went on to receive a diagnosis, and could contribute to the lack of clinical referrals for females. Please elaborate more on the implications of this in the second paragraph of the discussion.

Thank you for the insight – we have elaborated on your point in the Discussion on page 7.

I would reduce your discussion of the EMB theory. To truly show evidence of the EMB, you would want to compare TD boys to ASD girls, and the EMB would also suggest sex differences within ASD (since it’s just a shift in distribution).

The discussion of the EMB has been revised to be more cautious on page 7, and is currently 2 sentences.

Page 7 line 270: “These results do not support prominent speculations about ASD sex differences.” – I would temper this language. First, you’re only studying early childhood, where many studies find sex differences that present later. Second, you’re using tools that were designed to identify impairment (ADOS, VABS), not capture dimensional variability in symptoms. There still could be nuanced differences that aren’t be picked up on by these measures.

The sentence has been removed.

The paragraph that starts with “Across the past 20 years” feels unnecessary and like you are simply listing results from past studies.

The paragraph has been removed.

The discussion currently focuses on the lack of sex differences within the ASD sample. However, they fail to acknowledge the difference in sex ratio of their sample (22% of the ASD sample was female). Combined with the fact that they required participants to meet on the ADOS, it is possible that there is a group of girls who are showing mild symptoms but may truly have ASD or ASD-related concerns. I think it's worth highlighting this limitation or alternative explanation.

Thank you for raising this important point. We now include a thorough discussion on page 7 regarding the fact that some milder symptom ASD females likely were not included in the current sample. Regarding the male-to-female ratio, we now also mention that the expected male to female ratio is 3.8 to 1 based on the most recent CDC prevalence study¹², and the male to female ratio in the current study is slightly better at 3.6 to 1. Since an elevated male to female ratio is expected in ASD, and our study sample mirrors the US ASD population, we don't really consider the difference in sex ratio as a weakness. Rather, our sample seems to nicely match sample expectations.

Smaller points:

It appears that the authors required that participants meet cutoff on the ADOS. What is the rationale for this? The ADOS does not have perfect sensitivity, so we wouldn't expect all kids with ASD to meet.

As described in our recent publication²⁴, there is a 95% correlation between the psychologists judgement and ADOS diagnostic classification. However, you are correct, we do not exclusively rely on ADOS diagnostic but instead rely on clinician judgement, which takes into account information from a range of sources, including the ADOS. In approximately 5% of our sample, clinician judgement is different than the ADOS.

What do you mean by “ADOS-certified”? Were your assessors all research reliable on the ADOS?

Yes, all assessments were performed by Ph.D. level, ADOS research reliable psychologists. All psychologists at our Center are trained by an ADOS certified independent trainer and achieve reliability with that certified trainer. The criteria used for research reliable levels includes $\geq 85\%$ reliability on algorithm items and 100% reliability on the overall diagnostic classification.

Please see page 9 in the manuscript:

All assessments were administered by licensed clinical psychologists and eye tracking technicians blind to initial CSBS screen scores. To help ensure inter-clinician reliability, the lead clinical psychologist (CCB, ADOS-certified independent trainer) who has over 25 years of experience in toddler testing, was responsible for training other psychologists to achieve research reliable levels on the ADOS. ADOS reliability checks were conducted approximately 2x per year.

Pg. 3, line 113: Could you be more specific with the following sentence: “the 113 Communication and Symbolic Behavior Scales Infant-Toddler Checklist (CSBS)32, revealed poorer screen scores in ASD 114 boys than girls,”? I imagine you mean that boys demonstrated lower scores/more difficulties?

Please see page 4 in the manuscript:

Sex Differences in ASD. Although the parent-based screening tool used to recruit toddlers into the study, the Communication and Symbolic Behavior Scales Infant-Toddler Checklist (CSBS)²⁵, revealed lower screen scores in ASD boys than girls, we found no significant sex differences in 17 of the 18 standardized test scores presented in Figure 1.

Pg. 6, line 228 – you say that your sample was uniformly ascertained, but your methods describe that some kids came from clinical referrals (not ASD screening). I'd reduce how strongly you're claiming uniform ascertainment.

We now state that the 'majority' of our sample was uniformly ascertained instead of implying the entire sample.

References

- 1 Courchesne, E. *et al.* Embryonic origin of two ASD subtypes of social symptom severity: the larger the brain cortical organoid size, the more severe the social symptoms. *Mol Autism* **15**, 22 (2024). <https://doi.org/10.1186/s13229-024-00602-8>
- 2 Courchesne, E., Gazestani, V. H. & Lewis, N. E. Prenatal Origins of ASD: The When, What, and How of ASD Development. *Trends in neurosciences* **43**, 326-342 (2020). <https://doi.org/10.1016/j.tins.2020.03.005>
- 3 Bonnet-Brilhault, F. *et al.* Autism is a prenatal disorder: Evidence from late gestation brain overgrowth. *Autism Res* **11**, 1635-1642 (2018). <https://doi.org/10.1002/aur.2036>
- 4 Kolb, B., Harker, A. & Gibb, R. Principles of plasticity in the developing brain. *Dev Med Child Neurol* **59**, 1218-1223 (2017). <https://doi.org/10.1111/dmcn.13546>
- 5 Nelson, C. A., 3rd *et al.* Cognitive recovery in socially deprived young children: the Bucharest Early Intervention Project. *Science* **318**, 1937-1940 (2007). <https://doi.org/10.1126/science.1143921>
- 6 Guthrie, W. *et al.* The earlier the better: An RCT of treatment timing effects for toddlers on the autism spectrum. *Autism*, 13623613231159153 (2023). <https://doi.org/10.1177/13623613231159153>
- 7 Wetherby, A. & Prizant, B. (Paul H. Brookes, Baltimore, MD, 2002).
- 8 Wetherby, A. M. *et al.* Early indicators of autism spectrum disorders in the second year of life. *J Autism Dev Disord* **34**, 473-493 (2004).
- 9 Wetherby, A. M., Brosnan-Maddox, S., Peace, V. & Newton, L. Validation of the Infant-Toddler Checklist as a broadband screener for autism spectrum disorders from 9 to 24 months of age. *Autism* **12**, 487-511 (2008). <https://doi.org/10.1177/1362361308094501>
- 10 Pierce, K. *et al.* Detecting, studying, and treating autism early: The one-year well-baby check-up approach. *Journal of Pediatrics* **159**, 458-465.e456 (2011). <https://doi.org/10.1016/j.jpeds.2011.02.036>
- 11 Guthrie, W. *et al.* Accuracy of Autism Screening in a Large Pediatric Network. *Pediatrics* **144** (2019). <https://doi.org/10.1542/peds.2018-3963>
- 12 Maenner, M. J. *et al.* Prevalence and Characteristics of Autism Spectrum Disorder Among Children Aged 8 Years - Autism and Developmental Disabilities Monitoring Network, 11 Sites, United States, 2020. *MMWR Surveill Summ* **72**, 1-14 (2023). <https://doi.org/10.15585/mmwr.ss7202a1>
- 13 Riglin, L. *et al.* Variable Emergence of Autism Spectrum Disorder Symptoms From Childhood to Early Adulthood. *Am J Psychiatry* **178**, 752-760 (2021). <https://doi.org/10.1176/appi.ajp.2020.20071119>
- 14 Valicenti-McDermott, M., Rivelis, E., Seijo, R. & Shulman, L. Diagnosis of Autism in School Age and Adolescence in an Ethnically Diverse Population. *J Child Adolesc Psychopharmacol* (2024). <https://doi.org/10.1089/cap.2024.0004>
- 15 Leedham, A., Thompson, A. R., Smith, R. & Freeth, M. 'I was exhausted trying to figure it out': The experiences of females receiving an autism diagnosis in middle to late adulthood. *Autism* **24**, 135-146 (2020). <https://doi.org/10.1177/1362361319853442>
- 16 Stagg, S. D. & Belcher, H. Living with autism without knowing: receiving a diagnosis in later life. *Health Psychol Behav Med* **7**, 348-361 (2019). <https://doi.org/10.1080/21642850.2019.1684920>
- 17 Bölte, S. *et al.* Sex and gender in neurodevelopmental conditions. *Nature Reviews Neurology* **19**, 136-159 (2023). <https://doi.org/10.1038/s41582-023-00774-6>
- 18 Tooley, U. A., Bassett, D. S. & Mackey, A. P. Environmental influences on the pace of brain development. *Nat Rev Neurosci* **22**, 372-384 (2021). <https://doi.org/10.1038/s41583-021-00457-5>
- 19 Corbett, B. A. *et al.* Camouflaging in autism: Examining sex-based and compensatory models in social cognition and communication. *Autism Research* **14**, 127-142 (2021).
- 20 Conlon, O. *et al.* Gender differences in pragmatic communication in school-aged children with autism spectrum disorder (ASD). *Journal of Autism and Developmental Disorders* **49**, 1937-1948 (2019).

- 21 Kaat, A. J. *et al.* Sex differences in scores on standardized measures of autism symptoms: a multisite integrative data analysis. *Journal of Child Psychology and Psychiatry* **62**, 97-106 (2021).
- 22 Hiller, R. M., Young, R. L. & Weber, N. Sex differences in autism spectrum disorder based on DSM-5 criteria: evidence from clinician and teacher reporting. *Journal of abnormal child psychology* **42**, 1381-1393 (2014).
- 23 Zubizarreta, J. R., Paredes, R. D. & Rosenbaum, P. R. Matching for balance, pairing for heterogeneity in an observational study of the effectiveness of for-profit and not-for-profit high schools in Chile. *The Annals of Applied Statistics* **8**, 204-231, 228 (2014). <https://doi.org/https://doi.org/10.1214/13-AOAS713>
- 24 Pierce, K. *et al.* Evaluation of the diagnostic stability of the early autism spectrum disorder phenotype in the general population starting at 12 months. *JAMA pediatrics* **173**, 578-587 (2019).
- 25 Wetherby, A. M. & Prizant, B. M. *Communication and symbolic behavior scales: Developmental profile*. (Paul H Brookes Publishing Co., 2002).

REVIEWER COMMENTS:

Reviewer #1 (Remarks to the Author):

The authors have undertaken a very thorough response to prior reviews. I am satisfied with the response and modifications made and feel the current manuscript makes an important contribution to the literature

We appreciated your insightful comments and help in bringing this manuscript to such a high level of scholarship. Thank you!

Reviewer #2 (Remarks to the Author):

I appreciate the authors' careful consideration of the suggestions from reviewers, and believe the revised manuscript to be much improved. I have a few remaining concerns after reviewing the rebuttal and updated manuscript, as described below.

Additional justification for choosing 3 cluster rather than 2 is still needed. There should be some theoretical justification if there is no statistical reason to choose 3 over 2. Just wanting to look at "more nuanced" differences isn't sufficient of a reason. There will always be more nuanced differences if you are breaking a group up into 3 vs. 2 groups.

Thank you for raising this important point. We would like to provide additional clarification regarding our choice of 3 clusters rather than 2. While we agree that increasing the number of clusters can lead to more nuanced distinctions, we believe that in our case, the decision to use 3 clusters was supported both statistically and theoretically, as described below:

- **Statistical Justification:** As outlined in the manuscript, we evaluated 26 different clustering techniques, and the results indicated a tie between 2 and 3 cluster solutions. To resolve this, we proceeded with the 3-cluster solution, which was further validated through ANOVA and Silhouette score calculations. These statistical measures demonstrated that the 3-cluster solution provided better separation and clearer distinctions between the groups than a 2-cluster solution. Additionally, well-separated groupings, with meaningful differences across external validation variables such as ADOS RRB (as you suggested) were obtained.
- **Theoretical Justification – Capturing Clinical Heterogeneity:** In ASD research, it is well-documented that ASD is a highly heterogeneous condition, with significant variation in developmental abilities across social, language, and motor domains. By opting for 3 clusters, we were able to better capture this heterogeneity, identifying distinct groups of high, medium, and low performers. This stratification is particularly important in our study because it demonstrates that ASD toddlers are distributed across all three clusters, while TD toddlers were found predominantly in the high cluster, with a smaller percentage in the medium. This finding highlights key distinctions in the developmental profiles of these children, distinctions that would have been lost in a simpler 2-cluster analysis.
- **Clinical Relevance:** From a clinical perspective, distinguishing between toddlers with low, medium, and high abilities is critical for understanding the wide range of presentations within ASD. A 3-cluster model allows us to differentiate between ASD toddlers who may require varying levels of intervention, as some cluster more closely with typically developing toddlers in terms of their abilities. The medium cluster, in particular, captures this overlap between ASD and TD groups, providing clinically useful insights that a binary model would fail to reveal. Additionally, the separation of the TD group into only high and medium clusters reinforces the utility of the 3-cluster model in capturing the spectrum of abilities while reflecting the absence of low performance in typically developing toddlers.

I also noticed that it seems like you did not include the ADOS RRB score into your cluster analysis. If that is the case, I imagine that would be a nice “external variable” to compare the groups on.

Thank you for this suggestion. Based on your recommendation, we added RRB as an external validation variable and compared it across the 3 clusters. The results demonstrated a clear and statistically significant separation of the clusters based on the RRB score, further supporting the robustness of the 3-cluster solution.

Additional justification is needed in choices surrounding the ADOS. From the authors’ response, it seems like they compared algorithm raw score totals across modules and versions. This is simply not acceptable – the items included and scales are different. If you plan to compare across module, you need to use the CSS. If you want to look at raw scores, you should only look at differences within each version, module, and algorithm set (e.g., no words vs. some words).

Thank you for your feedback. We understand the concern regarding comparing raw ADOS scores across different modules and versions. While we acknowledge this, there are 2 key reasons why we deliberately selected to use the raw scores for this particular dataset, rather than the Calibrated Severity Scores (CSS).

First, the goal of our study is to examine the full range of ability (and disability) in ASD and to determine how this might differ between males and females. Unfortunately, the CSS compresses the score range and artificially removes variability, which is essential for detecting the subtle sex differences we are investigating. To illustrate, consider a hypothetical situation for 2 and 3-year-old toddlers that received Module 1 using the ‘Some Words’ algorithm. In our example, raw scores range between 22 and 23 for a sample of 7 females, while scores range between 25 and 28 for 7 males. In this case, we observe a significant difference in raw scores between the sexes (i.e., 22.43 vs 27, $t=9.05$; $p<.001$). However, if we convert these raw scores into CSS scores, both male and female scores would change to 10, eliminating the variability and masking the sex differences we aim to investigate. Figure 1.

Figure 1. Calibrated Symptom Severity Conversion Table (Left) and Hypothetical Dataset (Right).

Second, as noted in Wiggins et al. (2019), there is a high correlation ($r = 0.87$) between CSS and ADOS total raw scores, suggesting that the overall trends would remain consistent whether we used raw scores or CSS. Additionally, in our dataset the female-to-male ratio is similar across the modules (e.g., Toddler: 0.4, Module 1: 0.3, Module 2: 0.5), meaning that any differences in symptom severity between modules are unlikely to cause bias within either sex when using raw scores. Finally, the CSS was developed by Lord as a metric that is relatively independent of participant characteristics such as language ability and to create a metric suitable for tracking changes across development, goals that are not central for our study.

I also would strongly advocate against using ADOS scores in the TD group. If you use CSSs as suggested, those were only normed on diagnosed individuals and are not meant to be used in TD groups. Furthermore, there is recent evidence that the ADOS SA factor does not fit well in TD individuals, particularly females (Burrows et al., 2022), highlighting that it is not capturing meaningful variability in TD kids. That paper suggested not using ADOS scores as a dimensional measure for TD kids.

Regarding the use of ADOS scores in the TD group, we appreciate your suggestion to use calibrated CSS and the point that they are not normed for TD children. However, our primary aim was to explore subtle differences in social affect and restricted, repetitive behaviors that may still be present in the TD group, even though these measures are typically used for diagnosing ASD. Additionally, it is critical for our study to include a contrast group, such as TD children, to compare against the ASD group. Removing the TD group would not only make it impossible to examine this comparison, but it would also necessitate redoing the entire analysis, essentially transforming the study into a completely new paper. Given the scope and purpose of our analysis, redoing the entire study is not feasible.

While we acknowledge the point that ADOS was not originally designed for TD children, it's important to note that typical toddlers were included during the development and validation of the ADOS from the very beginning (Lord et al., 1989). The diagnostic items were rigorously tested and refined against both typical and non-ASD delayed children to ensure that they effectively indexed autism behaviors. Items that did not differentiate well between groups were revised, replaced, or dropped, leading to the current version of ADOS, which has been validated against typical and non-ASD delayed children to optimize its diagnostic accuracy. Lord et al work demonstrates how typical children were foundational in the development process to ensure the ADOS accurately identified autism specific behaviors.

Did you have the same proportion of ASD males and females in the train and test set?

Yes, the proportions of ASD males and females were similar across both the train (80%) and test (20%) datasets.

Train: 206/686 (f/m) = 0.3

Test: 48/176 (f/m) = 0.27

Please include the means and SD for each combination of M/F and ASD/TD for your primary scales of interest. Right now I'm only seeing the test statistic of the differences.

Please see Supplementary Table 16 in the manuscript.

Smaller points:

A smaller ADOS point, but in the manuscript, you describe that you used the ADOS-2, but in the response it says that you had a combination of ADOS-G and ADOS-2. Please check the manuscript for accuracy.

Please see page 8 in the manuscript:

Clinical Testing. Toddlers and their parents participated in a series of tests including the ADOS-2⁴² or ADOS-G, the MSEL⁷⁰, the Vineland-3⁴¹, the MacArthur-Bates CDI⁴³, CSBS³⁹, and GeoPref eye tracking test.

My understanding is that you should never report a p-value of 0.000. It should always say $p < 0.0001$. I defer to the journal editor on this point, but that's the convention I'm aware of.

P = 0.000 changed to $p < .001$.

Please reword the following caption for figure 1. It is currently unclear to me whether males or females had higher scores. It's also hard when the direction of effect (e.g., higher scores equals more impairment vs. higher scores equals

better skills in that area) is different across measures. Is there a way to put them on the same scale? “asterisks (*) indicate statistically significant sex differences, with girls outperforming boys in those marked areas.”

Thank you for your point. Both the ASD and TD columns in Figure 1 are presented on the same scale, and any bar crossing the red line corresponds to a p-value of less than 0.05. The length of the bars represents the strength of the statistical significance, with longer bars indicating lower p-values.

Regarding your concern about the direction of effect, we used the term "outperformance" to address the fact that in different measures, higher or lower scores may have different interpretations. For example, in the ADOS, a lower score represents better performance (i.e., less impairment), so girls outperform boys by having lower scores. In contrast, in the Vineland, higher scores indicate better abilities, so girls outperform boys by having higher scores. In all statistically significant sex differences, indicated by an asterisk (*) and crossing the red line, girls outperformed boys.

Please see page 16 in the manuscript

Figure 1. Comparison of sex differences in ASD and TD groups across various test subscales. Bars represent performance differences between males and females, with longer bars indicating stronger statistical significance (lower p-values). Any bar crossing the red line has a p-value less than 0.05. Asterisks () denote significant sex differences. In all cases where a significant difference is marked, girls outperformed boys. For the ADOS, where lower scores indicate less impairment, outperformance means lower scores for girls. For the Vineland, where higher scores represent better abilities, outperformance means higher scores for girls.*

We greatly appreciate the considerable time and effort you contributed to this manuscript. It is overall much improved thanks to your insightful comments. Thank you!

References

- 1- Wiggins, L. D., Barger, B., Moody, E., Soke, G., Pandey, J., & Levy, S. (2019). Brief report: the ADOS calibrated severity score best measures autism diagnostic symptom severity in pre-school children. *Journal of autism and developmental disorders*, 49, 2999-3006.
- 2- Lord, C; Rutter, M; Goode, S; Heemsbergen, J; Jordan, H; Mawhood, L; Schopler, R (1989). "Autism diagnostic observation schedule: A standardized observation of communicative and social behavior". *Journal of Autism and Developmental Disorders*. 19 (2): 185–212.